# Retrofitting Building Envelope Using Phase Change Materials and Aerogel Render for Adaptation to Extreme Heatwave: A Multi-Objective Analysis Considering Heat Stress, Energy, Environment, and Cost

Dileep Kumar [1,2], Morshed Alam [1,*] and Jay G. Sanjayan [1]

1   Centre for Sustainable Infrastructure and Digital Construction, Department of Civil and Construction Engineering, Swinburne University of Technology, Hawthorn 3122, Australia; dileepkumar@swin.edu.au (D.K.); jsanjayan@swin.edu.au (J.G.S.)
2   Department of Mechanical Engineering, Shaheed Zulfiqar Ali Bhutto Campus, Mehran University of Engineering and Technology, Khairpur Mir's 76062, Pakistan
*   Correspondence: mmalam@swin.edu.au

**Abstract:** Energy retrofitting the existing building stock is crucial to reduce thermal discomfort, energy consumption, and carbon emissions. However, insulating and enhancing the thermal mass of an existing building wall using traditional methods is a very challenging and expensive task. There is a need to develop a material that can be applied easily in an existing occupied building without much interruption to occupants' daily life while also having high thermal resistance and heat storage capacity. This study aimed to investigate a potential building wall retrofit strategy combining aerogel render and Phase change materials (PCM) because aerogel render is highly resistive to heat and PCM has high thermal mass. While a number of studies investigated the thermal and energy-saving performances of aerogel render and PCM separately, no study has been done on the thermal and energy-saving performance of the combination of PCM and aerogel render. In this study, the performance of 12 different retrofit strategies, including aerogel and PCM, were evaluated numerically in terms of heat stress, energy savings, peak cooling, emission, and lifecycle cost using a typical single-story Australian house. The results showed that applying aerogel render and PCM on the outer side of the external walls and PCM and insulation in ceilings is the best option considering all performance indicators and ease of application. Compared to the baseline, this strategy reduced severe discomfort hours by 82% in a free-running building. In an air-conditioned building, it also decreased energy use, peak cooling demand, $CO_2$ emission, and operational energy cost by 40%, 65%, 64%, and 35%, respectively. Although the lifecycle cost savings for this strategy were lower than the "insulated ceiling and rendered wall without PCM" case, the former one was considered the best option for its superior energy, emission, and comfort performance. Parametric analysis showed that 0.025 m is the optimum thickness for both PCM and aerogel render, and the 25 °C melting point PCM was optimum to achieve the best results amongst all performance indicators for a typical Australian house in Melbourne climate.

**Keywords:** building energy retrofitting; phase change materials; aerogel render; heat stress risk; energy savings; emission; lifecycle cost; peak cooling load

## 1. Introduction

The building sector consumes around 30% of total primary energy globally, which is expected to escalate up to 50% by 2050 due to population growth, human lifestyle changes, new technologies, and climate change [1]. Currently, fossil fuels are used to meet around 80% of the world's energy demand, which has an adverse social, economic, and environmental impact [2]. Therefore, the use of renewable energy sources and the

adoption of sustainable practices in buildings to minimize fossil fuel consumption are being investigated extensively around the globe [3].

Heat transfer through building envelopes (walls, roofs, windows, and doors) accounts for up to 60% of total heat loss and gain [4], which can be reduced by having insulated building envelopes [3], double and triple glazed windows [5], and thermochromatic windows [6]. A recent review of present authors compared thermal properties and performances of various building insulation materials [7]. It was concluded that a highly insulated building envelope significantly reduces total heating and cooling energy consumption and improves winter thermal comfort in a passive building. However, it resulted in overheating and increased peak cooling demand in a lightweight structure during a heatwave period because of the low heat storage capacity of the insulation and lightweight building materials. Therefore, the building envelope should have higher heat resistance and higher heat storage capacity to reduce heating and cooling energy use in an air-conditioned building and to improve thermal comfort in a passive building [7].

Like many other metropolitan cities, a significant percentage of the residential building stock in Melbourne, Australia were constructed before the introduction of the mandatory five stars (maximum is 10 stars) energy efficiency standard in 2005. In Victoria, approximately 86% of the currently occupied houses were built before 2005 with an average energy efficiency rating of only 1.81, which is very low [8]. Therefore, retrofitting those existing energy inefficient buildings is crucial to reduce energy consumption and harmful greenhouse gas emission from this sector. While insulating a wall in a new dwelling is straightforward, it is more challenging to insulate an existing occupied house unless it is under major renovation where claddings and plasterboards are removed. Sustainability Victoria [9] trialed the pump-in cavity wall insulation method, which resulted in 15.5% energy savings. However, this method is very expensive, and the average payback period was reported to be 29 years. Also, increasing the thermal mass of an existing building using traditional materials (such as bricks and concrete) is impossible. Hence, there is a need to develop a material that can be applied easily in an existing occupied building without much interference and has low thermal conductivity and high thermal storage.

Aerogel-based thermal insulating renders are introduced in the European Union market to insulate existing walls as an alternative to plasterboard and insulation panel [10]. It can be applied easily on the building envelope with limited impacts and interruptions on the occupants' daily life and building functionality [11]. It is a lightweight material with density and thermal conductivity of 150–220 kg/m$^3$ and 0.024–0.027 W/mK, respectively [11–13], depending on the percentage of aerogel granules in the mixture [14]. Aerogel render has higher compressive strength [15,16], low water permeability [17], and is inert to flame. However, aerogel's major drawback is the lower heat storage capacity compared to the conventional insulators and construction materials, as shown in Figure 1 [7]. The lower heat storage capacity causes high indoor temperature fluctuation and summer overheating that have adverse health impacts specifically for older occupants and infants. Therefore, there is a need to improve the heat storage capacity of the aerogel render integrated into the building envelope.

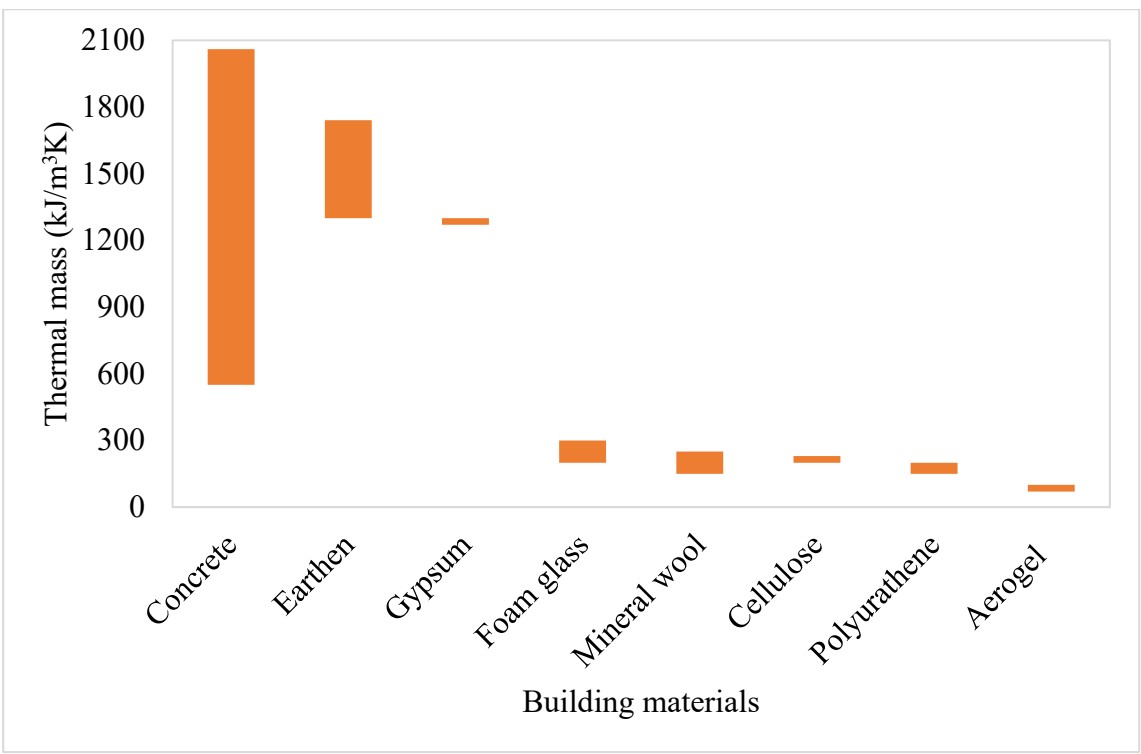

**Figure 1.** Thermal mass of building construction and insulation materials [7].

Integration of Phase change materials (PCM) in building envelope has been shown to increase heat storage capacity significantly [18]. Previous studies successfully integrated micro-encapsulated PCM in building materials, including structural materials [19,20], plaster and mortar [21,22] and insulation [23,24] to improve heat storage capacity and resulting in energy savings in air-conditioned buildings and improving summertime thermal comfort in passive buildings [25]. Hasnat et al. [26] reported a 34% reduction in thermal discomfort hours through the installation of Bio-PCM pouches in the ceilings of a Melbourne house. In other studies, the use of PCM-enhanced geopolymer coating and cement mortar reduced the test hut indoor air temperature up to 2.8 °C [25] and 2.4 °C [27], respectively, in summer, compared to an identical hut containing ordinary cement plaster. Cui et al. [28] developed a thermal energy storage concrete (TESC) using macro-encapsulated lauryl alcohol-lightweight aggregate PCM. The maximum air temperature in a test room (500 mm × 500 mm × 500 mm) containing TESC in the wall and roof was up to 9 °C and 5 °C lower, respectively, compared to no PCM room. Piti et al. [20] incorporated 7.8% Polyethylene glycol type 1450 by weight into a lightweight concrete that increased the heat storage capacity of concrete from 0.92 to 7.7 kJ/kg. Kosny et al. [29] reported that PCM-blended cellulose insulation has similar thermal insulating properties to cellulose insulation up to 30% of PCM addition with a staggering increment in heat storage capacity from 1.04 J/g to 60–80 J/g. The application not only reduced the cooling load (35–40%) but also decreased the heating demand up to 16% in a conventional house located in southern California. Rathore et al. [30] found that the PCM-embedded concrete panel reduces summertime thermal amplitude and time lag by 40.67–59.79% and 7.19–9.18%, respectively, benefitting with cooling energy savings of 0.40 US $/day.

The literature review shows that a PCM combined with aerogel render would be an ideal candidate to retrofit existing buildings because of its ease of application, lower thermal conductivity, and higher thermal mass. While a number of studies investigated the thermal and energy-saving performances of aerogel render and PCM separately, no study has been done on the thermal and energy-saving performance of PCM combined with aerogel render. A multi-objective optimization study could provide important information

regarding the optimum PCM melting temperature and optimum PCM and aerogel layer thickness to achieve the desired comfort, energy, environment, and cost performance of a retrofitted building [31–33].

Therefore, this study aims to investigate the performance of building envelope retrofitted with PCM combined with aerogel render in terms of heat stress, energy savings, peak cooling, emission, and lifecycle cost. The specific objectives are:

(1) To investigate and identify the best retrofit combinations using PCM blanket, aerogel render, and insulation in passive and air-conditioned buildings.
(2) To determine the optimum PCM temperature, PCM thickness, and aerogel render thickness for the identified best retrofit combination.

In this paper, Section 2 describes the research methodology, including simulation information, metrological parameters (ambient temperature, relative humidity, solar radiation density, and wind speed), material characteristics, and models. It also describes the case study building and proposed retrofit strategies, methods of thermal discomfort assessment, energy use estimation, emission calculations, and lifecycle cost analysis. The comparative analysis of results for different retrofit strategies is shown in Section 3. In Section 4, the study results are discussed further, and the best PCM-aerogel combination is proposed along with optimum phase change temperature and thickness, considering all performance criteria. Finally, Section 5 presents the concluding remarks and future directions.

## 2. Methodology

### 2.1. Case Study Building Description

A typical single-story Australian house was used as a case study to investigate the thermal performance of retrofit strategies. The selected case study building is one of the eight representative Australian houses used to develop the nationwide house energy rating system (NatHERS) in Australia [34]. According to the Australian Building Code Board, the selected single-story house model is one of the two most typical representations of single-story detached houses in Australia. Approximately 72.9% of the Australian dwellings fall in the category of single-story detached houses [35]. The studied house is a four-bedroom, two-bathroom family house with a floor area of 232 m$^2$. Figure 2 [36,37] shows the isometric view and thermal zones of the simulated house, along with the orientation from the north. The thermophysical properties of building materials are given in Table 1. The construction of the building envelopes varies depending on the simulation cases and is presented in the following section.

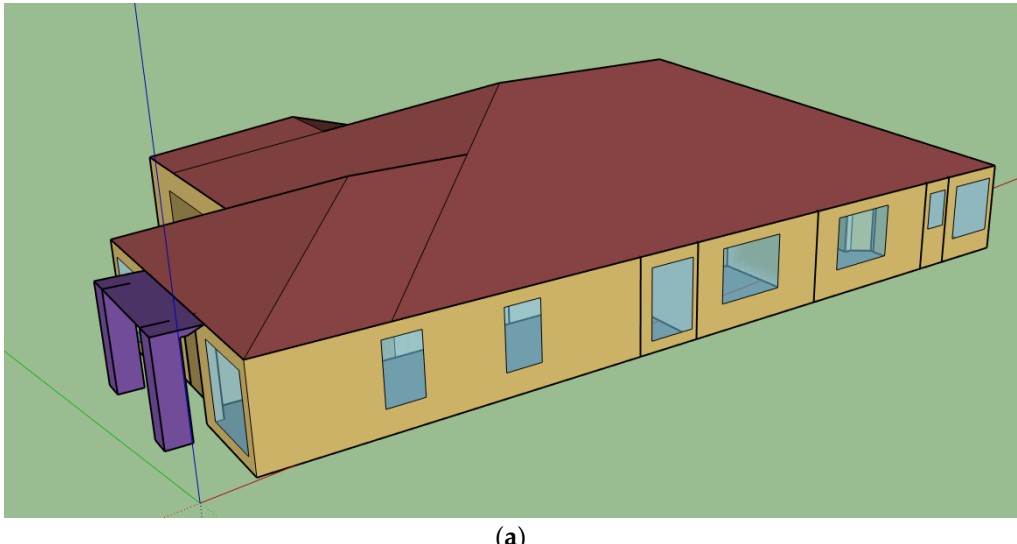

(a)

**Figure 2.** *Cont.*

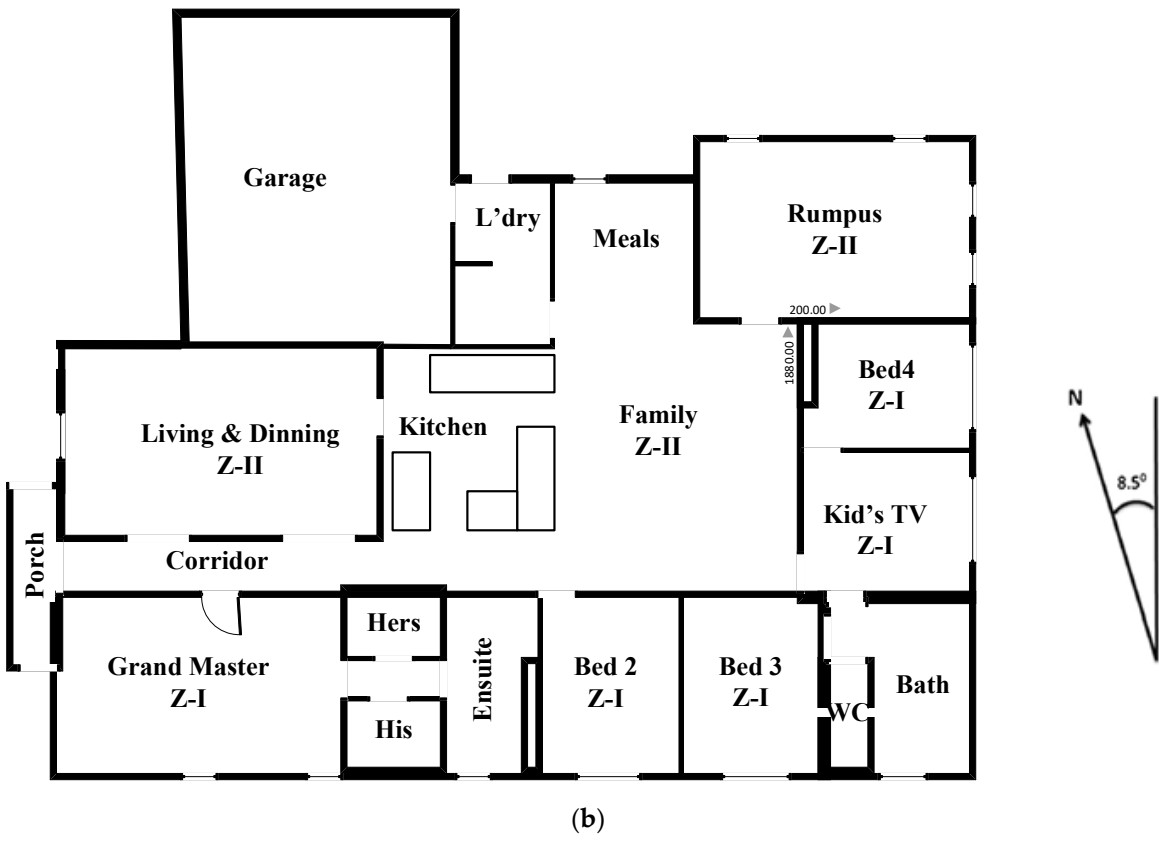

(**b**)

**Figure 2.** The simulated single-story house: (**a**) Isometric view; (**b**) Thermal zones in 2D.

**Table 1.** Thermo-physical properties of building materials.

| Building Materials | Thermo-Physical Properties | | | |
|---|---|---|---|---|
| | Thickness (m) | Conductivity (W/m K) | Density (kg/m$^3$) | Specific Heat (J/kg K) |
| Concrete | 0.100 | 1.42 | 2400 | 880 |
| Brick veneer | 0.110 | 0.61 | 1690 | 878 |
| Roof insulation | 0.044 | 0.044 | 12 | 883 |
| Roof tiles | 0.02 | 1.42 | 2400 | 880 |
| plasterboard | 0.013 | 0.17 | 847 | 1090 |
| Carpet | 0.02 | 0.0465 | 104 | 1420 |
| Timber doors | 0.05 | 0.16 | 1122 | 1260 |
| PCM | See Table 4 | 0.2 | 235 | 2400 |
| Aerogel render [36,37] | 0.02 | 0.024 | 100 | 1000 |

## 2.2. Building Energy and Thermal Simulations

The case study building was simulated using building simulation software EnergyPlus v9.2. EnergyPlus v9.2 was developed by the U.S. Department of Energy (DOE). Google Sketchup provided a comprehensive and powerful graphical user interface to EnergyPlus. The simulations were carried out considering the weather file for the year 2009 (the Bureau of Meteorology, Government of Australia), which reported an extreme heatwave in January, as seen in Figure 3 [38]. Melbourne exhibits a temperate oceanic climate, which has a high diurnal temperature swing. Temperate climatic zones are advantageous for PCM application because it allows complete melting/freezing cycle during summer and improves summertime thermal comfort.

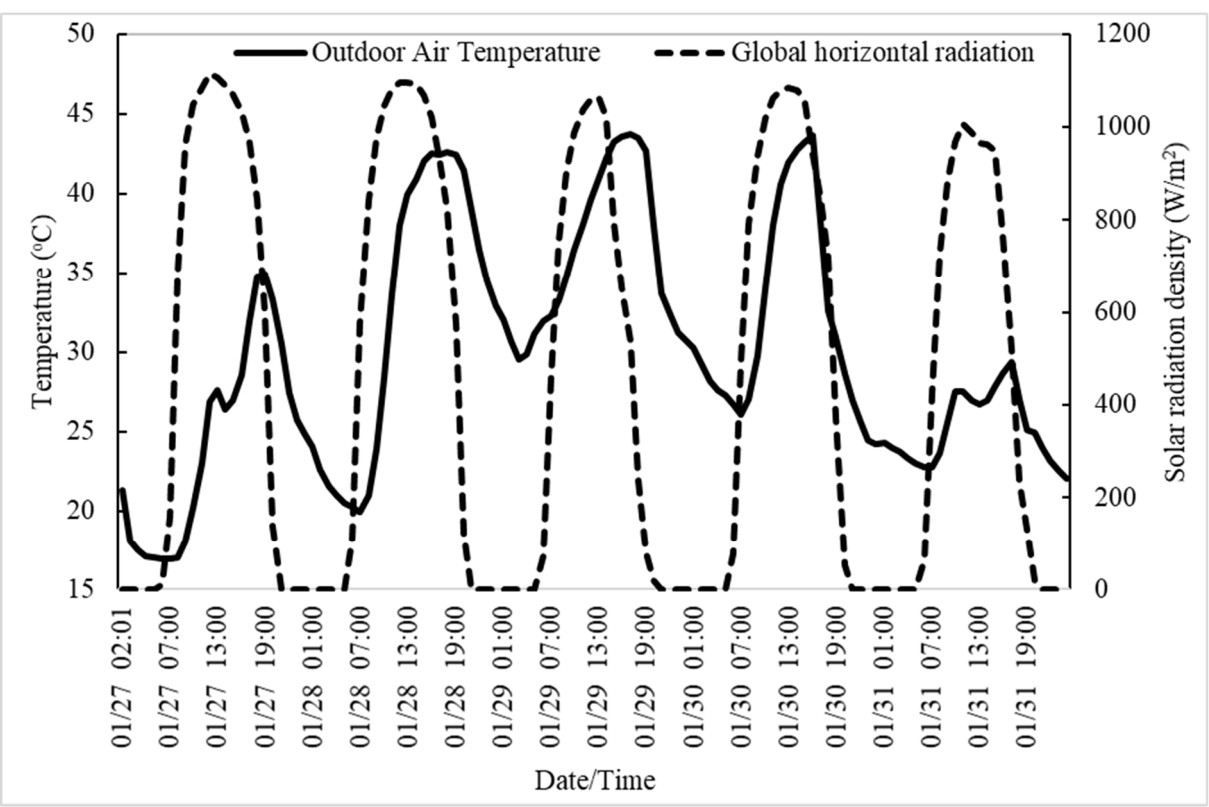

**Figure 3.** Climatic conditions during extreme heatwave period (27–31 January 2009 Melbourne).

The simulations were run using a conduction finite-difference algorithm (ConFD), which allows the simulation of temperature-dependent properties of PCM. In addition, this study used a fully implicit CondFD scheme for dynamic thermal simulation. This scheme accounts for time-dependent phase change phenomenon of PCM using the enthalpy-temperature function.

$$C\rho\Delta x\frac{T_{b+1}^{a+1} - T_b^a}{\Delta x} = \left\{ k_i \frac{\left( T_{b+1}^{a+1} - T_b^{a+1} \right)}{\Delta x} + k_j \frac{\left( T_{a-1}^{a+1} - T_b^{a+1} \right)}{\Delta x} \right\} \tag{1}$$

The specific heat capacity of PCM is temperature-dependent and is updated at every iteration in EnergyPlus according to Equation (2). The effective specific heat capacity of PCM is calculated as:

$$C = \frac{h_i^j - h_i^{j-1}}{T_i^j - T_i^{j-1}} \tag{2}$$

where;

$C$ = specific heat capacity of material (kJ/kg K)
$\rho$ = density of material (kg/m$^3$)
$h$ = specific enthalpy (kJ/kg)
$T$ = Temperature (°C)
$b$ = temperature node, $b-1$ and $b+1$ are adjacent inner and outer nodes.
$a+1$ and $a$ = simulation time and previous time step
$k_i$ and $k_j$ = material's thermal conductivity at a different node.

This study includes BioPCMs having melting point temperature ranges between 20 °C and 32 °C. The thermophysical properties of PCMs are presented in Table 1. Figure 4 shows the enthalpy-temperature graphs of PCMs with different phase change temperatures used in this study [39]. Each PCM has a phase transition range of 4 °C. For example,

PCM24 means it will complete a phase change cycle between 22 °C and 26 °C. While the individual thermal properties of PCM and aerogel layers are known, the thermal properties of PCM-integrated aerogel render are not yet known. The ultimate goal of this project is to develop a PCM-integrated aerogel render for easy retrofitting of an existing building wall. This simulation study was carried out as part of the feasibility study to know how the combination of PCM and aerogel influences building thermal performance and energy consumption. Therefore, for the purpose of this feasibility study and for the sake of simplicity, we assumed PCM and aerogel render as separate layers in this simulation study.

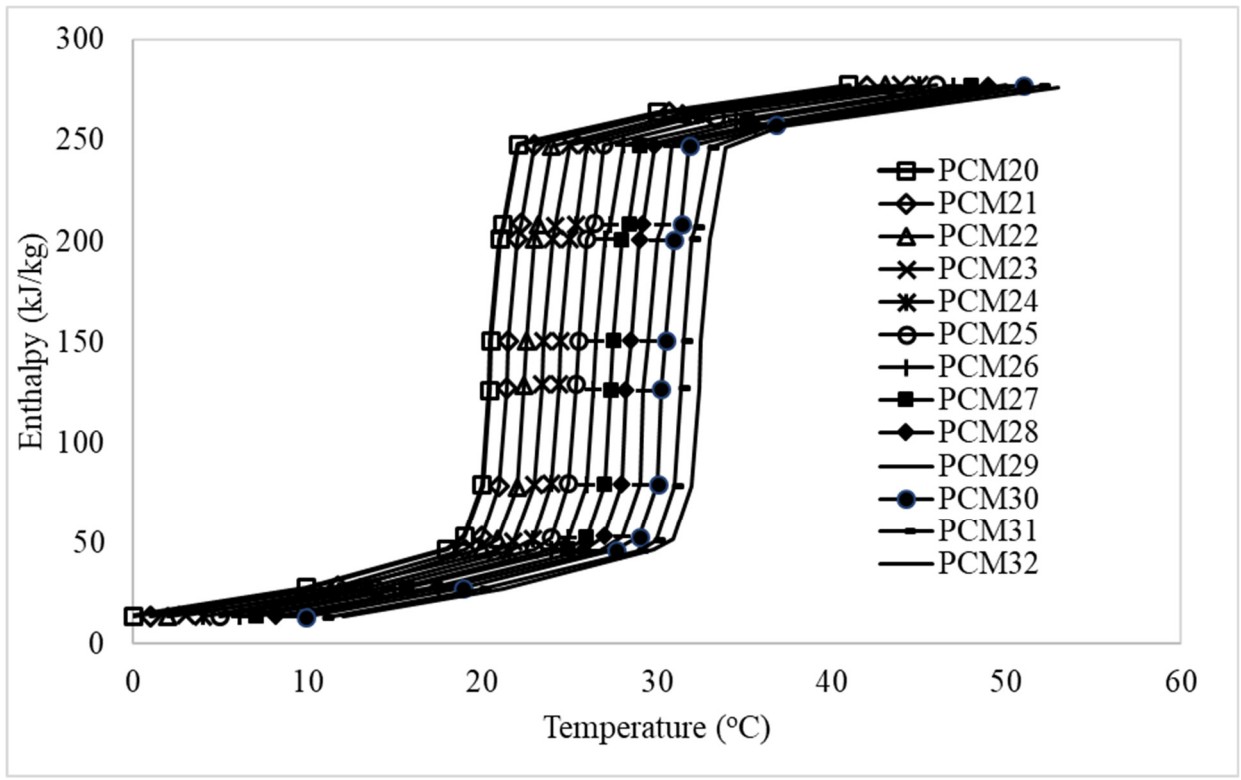

**Figure 4.** Enthalpy-temperature curve of BioPCM [2].

The house was assumed to be occupied by four residents, with different schedules for weekdays and weekends [40]. Figure 5 exhibits the activity level of occupants in different house zones. The GroundHeatTransfer: Slab module of the EnergyPlus software was used to simulate the ground source heat transfer [2]. Moreover, an Effective leakage area model was used to simulate infiltration [41].

Each simulation was conducted twice, considering different retrofit strategies: (1) with and (2) without an HVAC system. The simulations with the HVAC system were used to evaluate the impact of retrofit strategies on total annual energy and peak cooling demand. The simulations without HVAC were used to assess the impact of retrofit strategies on indoor heat stress during a heatwave. The risk of a power outage is very high during the hot summer period, which may leave the HVAC system out of order and pose a significant threat of heat stress to the occupant. Therefore, the retrofit strategies need to be evaluated for both HVAC and no HVAC scenarios. Table 2 gives information about operating, economic parameters, and their respective references.

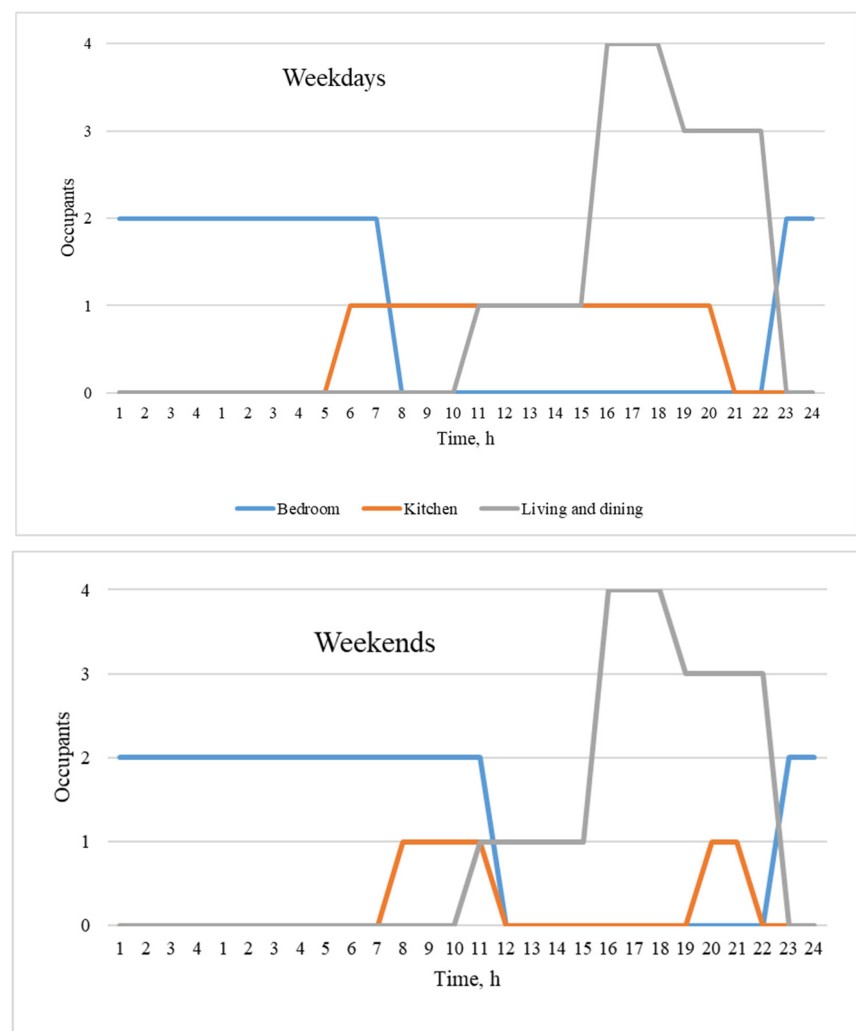

**Figure 5.** Occupants' schedule in a typical Australian house [40].

**Table 2.** Assumption of different operating conditions and economic parameters.

| Operating Conditions | Value | Standard |
|---|---|---|
| Time step | 3 min | Tabares-Velasco et al. [42] |
| Thermostat setpoints | | The house energy rating standards of Australia [43] |
| Heating (°C) | 20 | (00:00–8:00 and 16:00–24:00 h) |
| Cooling (°C) | 24 | |
| People (person) | 4 | |
| Metabolic rate (W/person) | | ASHRAE [44] |
| Writing, seating, standing | 108 | |
| Cooking, cleaning | 171 | See Figure 5 |
| Reading, relaxing | 108 | |
| Lighting (W/m$^2$) | 2.5 | Australian building code boards [45] |
| Electric equipment (W/m$^2$) | 1.875 | |
| Economic Parameter | | |
| Ceiling insulation | 5.93 AUD/m$^2$ | [46] |
| PCM | 4.33 AUD/kg | [47] |
| Aerogel render (AG) | 50–62 AUD/kg | ENERSEN, France [14] |
| Electricity usage rates | 0.31 AUD/kWh | |
| Electricity supply charges | 1.1408 AUD/day | |
| Natural gas usage rate | 0.115 AUD/kWh | Energy Australia [48] |
| Natural gas supply charges | 0.759 AUD/day | |

**Table 2.** *Cont.*

| Operating Conditions | Value | Standard |
|---|---|---|
| Electricity Emission Factor | 1.08 kgCO$_{2\text{-eq}}$/kWh | Australian national greenhouse accounts [49] |
| Natural gas emission factor | 3.9 kgCO$_{2\text{-eq}}$/GJ | |
| Ducted cooling system (COP$_{eq}$) | 1.96 | [50] |
| Ducted gas heating system ($\eta$) | 52.5% | [51] |
| Conversion Factor: Electricity | $3.6 \times 10^6$ J/kWh | [52] |
| Heating Value natural gas | $34.526 \times 10^6$ J/m$^3$ | [52] |
| Inflation rate ($i$) | 1.93% | |
| Interest rate ($d$) | 6% | Office of Best Practice Regulation [53] |
| Lifetime (LT) | 40 years | Australian building code boards [45] |

Nosrati and Berardi [17] showed that aerogel render thermal conductivity changes with ambient air relative humidity. To consider this moisture dependency, moisture dependent thermal conductivity data of 90% Aerogel-enhanced plaster data, as reported in [17], were used to vary the thermal conductivity of aerogel in the simulation using energy management system (EMS) object in EnergyPlus.

Figure 6 exhibits a negligible difference in heating and cooling load for hygrothermal and non-hygrothermal simulation because the annual average relative humidity of Melbourne is only 55%, which meagerly changes aerogel render thermal conductivity. The average monthly relative humidity varies from a minimum of 48% in January (Summer) to 72% in June. On the other hand, the relative humidity of around 95% has been shown to impact thermal conductivity significantly [17]. Therefore, in this study, aerogel render was simulated without considering their hygrothermal properties.

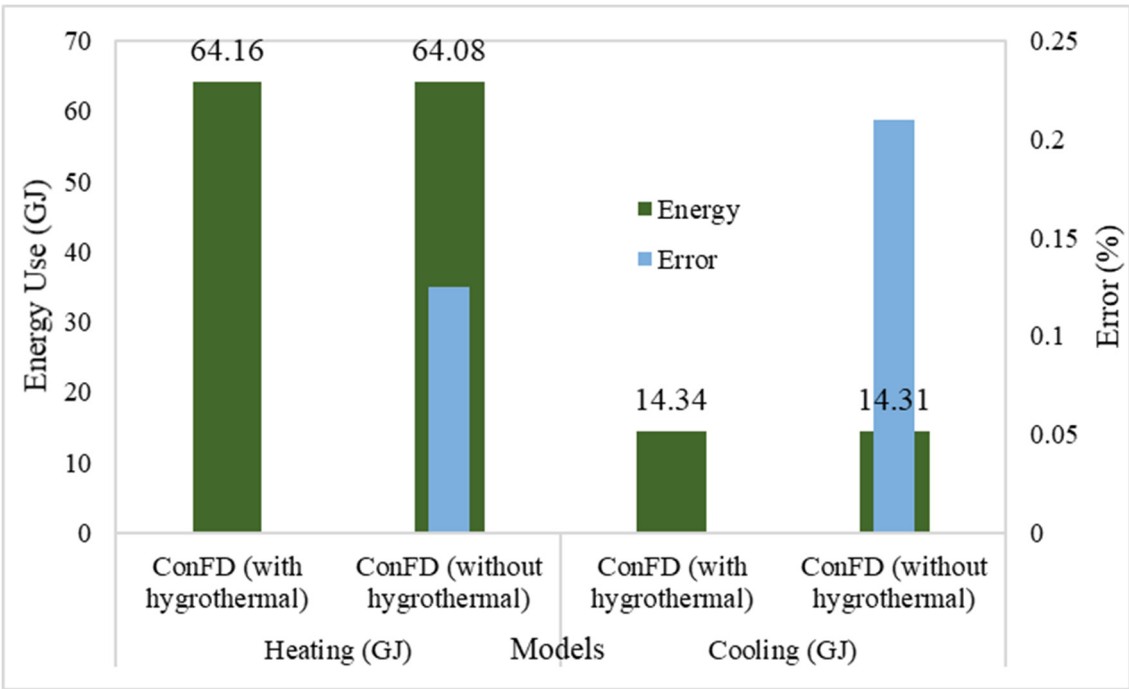

**Figure 6.** Comparison of heating and cooling energy use in building in case of hygrothermal and non-hygrothermal simulation.

Validation of the numerical model is a pre-condition for any simulation-based study. Several studies were conducted by many researchers and the EnergyPlus developer team to validate the EnergyPlus PCM simulation algorithm using analytical (Stefan Problems) [42], comparative testing [54], and field studies [26,55] approaches. For instance, Tabares-Velasco et al. [42] suggested that EnergyPlus is a reliable tool for simulating PCM by

comparing the simulation outcomes with experimental investigations. They recommended that the simulation should be conducted considering time step ($\leq$3 min). Moreover, the present author developed and validated the EnergyPlus model of a real duplex house with PCM in a previous study [26]. The single-story house model that was used in this study was developed using a modeling approach similar to that validated duplex house model. This single-story house model was also successfully used to evaluate heat stress conditions in a previous study of the present authors [56]. Therefore, the use of a validated modeling approach provides it with secondary validation.

### 2.3. Benchmark Studies

Table 3 shows the simulation cases with different retrofit strategies. Construction details of the ceiling, internal wall, and external wall are illustrated in Figure 7. Each simulation case was assessed considering heat stress risks, energy-saving potential, emissions, and lifecycle costs. Case 1 is the baseline house without any insulation in the ceiling and walls because this study aims to investigate the retrofitting potential of existing energy-inefficient building stock combining aerogel render with PCM. As mentioned in Section 1, the energy efficiency rating of a significant percentage of existing occupied houses in Melbourne is very low. Therefore, the selection of a baseline case without any insulation in the ceiling and walls is justified.

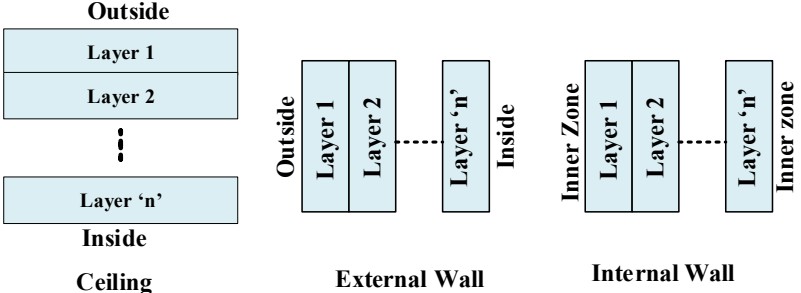

**Figure 7.** Schematics of envelope constructions in different simulation cases.

To allow direct comparison between different cases with PCM, the total amount (kg) of PCM was kept constant in the simulation cases with PCM. This was done by varying the thickness of the PCM layer according to the application surface area.

### 2.4. Parametric Studies

The parametric analysis determined the optimum phase change temperature (OPCT) for each retrofit case with PCMs. PCMs with phase change temperatures ranging from 18 °C to 32 °C were considered, as shown in Figure 4. After the selection of the best retrofit combinations, the second set of parametric studies were carried out to identify optimum aerogel render and PCM thickness. JePlus v2.1 was used together with EnergyPlus to conduct the parametric analysis by varying aerogel render thickness from 0.01 m to 0.05 m and the PCM thickness from 0.005 m to 0.025 m.

**Table 3.** Construction details of the simulation cases.

| Retrofit Cases | Description | Ceiling | | | External Walls | | | | Internal Walls | | | | | | |
|---|---|---|---|---|---|---|---|---|---|---|---|---|---|---|---|
| | | Layer 1 | Layer 2 | Layer 3 | Layer 1 | Layer 2 | Layer 3 | Layer 4 | Layer 1 | Layer 2 | Layer 3 | Layer 4 | Layer 5 | Layer 6 | Layer 7 |
| Case 1 | Baseline (Uninsulated wall and ceiling) | Ceiling Plasterboard | | | Brick Veneer | Wall Plasterboard | | | Wall Plasterboard | Air gap | Wall Plasterboard | | | | |
| Case 2 | Rendered wall and uninsulated ceiling | Ceiling Plasterboard | | | Aerogel Render | Brick Veneer | Wall Plasterboard | | Wall Plasterboard | Air gap | Wall Plasterboard | | | | |
| Case 3 | Rendered wall and insulated ceiling | Insulation | Ceiling Plasterboard | | Aerogel Render | Brick Veneer | Wall Plasterboard | | Wall Plasterboard | Air gap | Wall Plasterboard | | | | |
| Case 4 | Rendered wall coupled with PCM and uninsulated ceiling | Ceiling Plasterboard | | | Aerogel Render | PCM | Brick Veneer | Wall Plasterboard | Wall Plasterboard | Air gap | Wall Plasterboard | | | | |
| Case 5 | Rendered wall coupled with PCM and uninsulated ceiling coupled with PCM | PCM | Ceiling Plasterboard | | Aerogel Render | PCM | Brick Veneer | Wall Plasterboard | Wall Plasterboard | Air gap | Wall Plasterboard | | | | |
| Case 6 | Rendered wall coupled with PCM and insulated ceiling | Insulation | Ceiling Plasterboard | | Aerogel Render | PCM | Brick Veneer | Wall Plasterboard | Wall Plasterboard | Air gap | Wall Plasterboard | | | | |
| Case 7 | Rendered wall coupled with PCM and insulated ceiling coupled with PCM | Insulation | PCM | Ceiling Plasterboard | Aerogel Render | PCM | Brick Veneer | Wall Plasterboard | Wall Plasterboard | Air gap | Wall Plasterboard | | | | |
| Case 8 | Rendered wall and uninsulated ceiling | Ceiling Plasterboard | | | Brick Veneer | Wall Plasterboard | Aerogel Render | | Aerogel Render | Wall Plasterboard | Air gap | Wall Plasterboard | Aerogel Render | | |
| Case 9 | Rendered wall and insulated ceiling | Insulation | Ceiling Plasterboard | | Brick Veneer | Wall Plasterboard | Aerogel Render | | Aerogel Render | Wall Plasterboard | Air gap | Wall Plasterboard | Aerogel Render | | |
| Case 10 | Rendered wall coupled with PCM and uninsulated ceiling | Ceiling Plasterboard | | | Brick Veneer | Wall Plasterboard | PCM | Aerogel Render | Aerogel Render | PCM | Wall Plasterboard | Air gap | Wall Plasterboard | PCM | Aerogel Render |
| Case 11 | Rendered wall coupled with PCM and uninsulated ceiling coupled with PCM | PCM | Ceiling Plasterboard | | Brick Veneer | Wall Plasterboard | PCM | Aerogel Render | Aerogel Render | PCM | Wall Plasterboard | Air gap | Wall Plasterboard | PCM | Aerogel Render |
| Case 12 | Rendered wall coupled with PCM and insulated ceiling | Insulation | Ceiling Plasterboard | | Brick Veneer | Wall Plasterboard | PCM | Aerogel Render | Aerogel Render | PCM | Wall Plasterboard | Air gap | Wall Plasterboard | PCM | Aerogel Render |
| Case 13 | Rendered wall coupled with PCM and insulated ceiling coupled with PCM | Insulation | PCM | Ceiling Plasterboard | Brick Veneer | Wall Plasterboard | PCM | Aerogel Render | Aerogel Render | PCM | Wall Plasterboard | Air gap | Wall Plasterboard | PCM | Aerogel Render |

Group descriptions: Cases 2–7 — "Applying render on the outer parts of the external wall". Cases 8–13 — "Applying render on the inner parts of the exterior and interior wall".

### 2.5. Analysis Methods

2.5.1. Indoor Heat Stress Risk and Thermal Discomfort

In this study, the thermal discomfort index was used for analyzing the heat stress risk in different retrofit cases. The thermal discomfort index (TDI) is the average of indoor air wet-bulb and dry-bulb temperature, which is estimated using Equation (3) [57].

$$TDI = \frac{T_{drybulb} + T_{wetbulb}}{2} \tag{3}$$

where, $Td$ and $Tw$ denote dry bulb and wet bulb temperature of indoor air. The EnergyPlus model by default calculates the dry-bulb temperature, relative humidity, and barometric pressure of each zone at every time step during the simulation. The wet bulb temperature was calculated using an advanced functionality of EnergyPlus known as EMS application, which uses dry-bulb temperature, relative humidity, and barometric pressure as input [58]. The heat stress risk can be classified as mild, moderate, and severe, observing the behavior of a large population group under different climates. Epstein and Moran established environmental heat stress criteria as tabulated in Table 4 [57].

**Table 4.** Threshold values of thermal discomfort hours [57].

| Discomfort Index (DI) | Classification of Heat Stress |
|---|---|
| DI < 22 | No heat stress is encountered. |
| 22 < DI < 24 | A mild sensation of heat stress. |
| 24 < DI < 28 | Moderate heat stress, people feel very hot, and physical work may be performed with some difficulties. |
| DI > 28 | Heat stress is severe; people engaged in physical work are at increased risk for heat exhaustion and heatstroke. |

2.5.2. Energy Savings

Energy savings (*ES*) is the measure of the percentage of energy consumption reduction in the retrofitted building. It was calculated by using Equation (4) [2].

$$ES = \left( \frac{EC_r - EC_{ret}}{EC_r} \right) \times 100\% \tag{4}$$

where $EC_r$ and $EC_{ret}$ denote energy consumption of reference and retrofitted buildings.

2.5.3. Emission Reduction

In Melbourne, almost 69% of households use the natural gas heater for heating, and 36% of households (highest among other cooling methods) use reverse cycle air-conditioning for cooling [59]. Therefore, this study assumed that the building is equipped with a natural gas heater and split air conditioner to meet the heating demand in winter and cooling demand in summer. The operational GHG emission is the product of heating and cooling energy use and their respective emission factors. Scope 1 emission factor (51.53 kg $CO_2$-e/GJ for Melbourne) was applied to heating demand, while scope 2 emission factor (1.07 kg $CO_2$-e/kWh for Melbourne) was considered for cooling demand [49]. Equation (5) was used to calculate the greenhouse gas emissions:

$$EE_{GHG} = EC_{el} \cdot EF_{el} + EC_{ng} \cdot EF_{ng} \tag{5}$$

where $EC_{el}$ and $EC_{ng}$ denote cooling and heating energy use and their respective emission factors are denoted by $EF_{el}$ and $EF_{ng}$, respectively. The percentage of emission reduction is calculated as.

$$ER = \left( \frac{EE_{GHG,r} - EE_{GHG,ret}}{EE_{GHG,r}} \right) \times 100\% \tag{6}$$

where $EE_{GHG, r}$ and $EE_{GHG, ret}$ denote *GHG* emission associated with energy use in reference and retrofitted buildings.

2.5.4. Lifecycle Cost Analysis

The lifecycle analysis considers the initial investments, operating and maintenance costs up to the disposal, and recovery costs. However, the economic optimization of building envelope walls and roofs excludes maintenance, renewal, and disposal costs. It only includes the investment of proposed alternative and operation energy costs. Operational energy cost varies according to interest and the inflation rate over the expected building lifetime, determined through the present worth factor (*PWF*) over a lifetime. The *PWF* is calculated using Equation (7) [3].

$$PWF = \sum_{j=1}^{LT} \frac{(1+i)^{j-1}}{(1+d)^j} = \left\{ \begin{array}{cc} \frac{1}{d-i}\left[1 - \left(\frac{1+i}{1+d}\right)^{LT}\right] & if \ d \neq i \\ \frac{LT}{1+i} & if \ d = i \end{array} \right\} \tag{7}$$

where, $i$, $d$, and $LT$ denote inflation rate, interest rate, and the lifetime of a building. The total lifecycle cost is the sum of the annual operation energy cost and the investment cost of retrofit strategies. Which is calculated as [3]:

$$LCC = C_e \cdot PWF + C_i \tag{8}$$

where $C_e$ and $C_i$ denote operation energy cost and initial investment, which are estimated as

$$C_e = \frac{ES_c \ C_{EL}}{COP \ LHV_{EL}} + \frac{ES_h \ C_{NG}}{\eta \ LHV_{NG}} \tag{9}$$

$$C_I = \sum_{i=o}^{n} C_{ins} + C_{render} + C_{PCM} \tag{10}$$

$ES_c$ and $ES_h$ are the cooling and heating energy savings; $C_{EL}$ and $C_{NG}$ are the unit cost of electricity and natural gas; $LHV_{EL}$ and $LVG_{NG}$ are the lower heating value of electricity and natural gas, respectively; $COP$ denotes the co-efficient of performance of non-ducted air conditioning unit; and $\eta$ represents the efficiency of heating system. Moreover, $C_{ins}$, $C_{render}$, and $C_{PCM}$ denote insulation cost, aerogel render cost, and *PCM* cost, respectively. In this study, a ducted cooling system with a *COP* of 2.79 with 30% duct losses (equivalent to overall *COP* of 1.96) was used, which is the minimum energy performance standard (AS/NZS 3823.2) for air-conditioners used in the Australian state [50].

The maximum cost-saving (*CS*) is the difference of lifecycle cost of reference (*LCC_ref*) and retrofitted (*LCC_ret*) envelope, which is estimated as:

$$CS = LCC_{ref} - LCC_{ret} \tag{11}$$

## 3. Results

### 3.1. Performance of the Retrofitting Strategies in Terms of Heat Stress Risk

The severe discomfort hours corresponding to optimum PCT in the living and bedroom 4 are presented in Figure 8. The living, family, rumpus, and kids tv zones have different occupancy schedules than bedrooms as shown in Figure 5. While discomfort hours were calculated for all zones of the house, results of only two zones were presented here for the sake of brevity. One zone from living type (mostly occupied during daytime) and one zone from bedrooms were selected for the presentation of results.

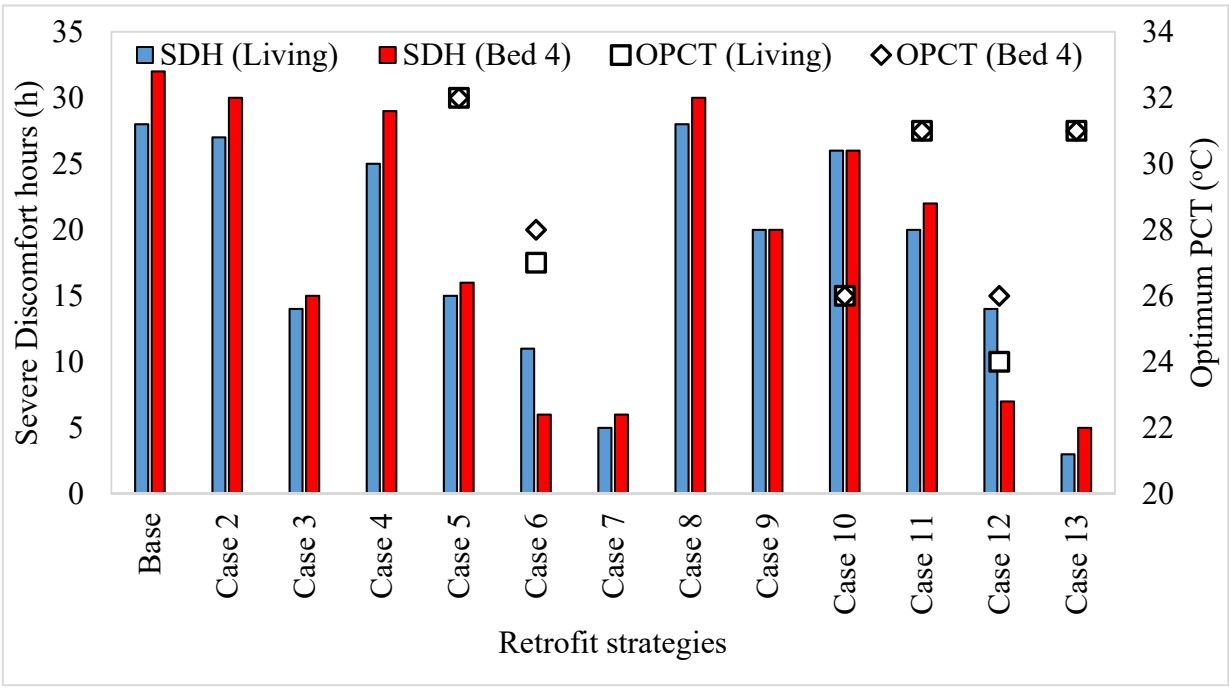

**Figure 8.** Severe discomfort hours corresponding to optimum phase change temperature in living and bedroom.

The discomfort hours in bedroom 4 were slightly higher than in the living room. It could be because of poor cross ventilation in the confined space of Bed 4, which is heated up by the radiation from the eastern sun in the morning. Also, it has a large window to wall ratio compared to the living room. In contrast, the living room has a large internal opening to the corridor, which results in higher cross ventilation. Nevertheless, both zones showed similar severe discomfort hours and optimum phase change temperature corresponding to the retrofit strategies.

Without PCM, the application of ceiling insulation and aerogel rendering on the outer part of the wall (Case 3) was the best combination to reduce discomfort hours. It performed better than the PCM combined wall and ceiling strategies without ceiling insulation (Case 4, 5, 10, and 11). Insulated ceiling mitigated heat stress risk better than the bared ceiling and aerogel rendered wall because heat transfer through the ceiling is higher than the wall [60] and hence, insulation of ceiling significantly reduces heat transfer through the roof than walls. Without ceiling insulation, other retrofitting measures in walls and ceilings were not very effective in minimizing the discomfort hours.

Moreover, in the presence of ceiling insulation, the external wall with aerogel render on its outer part (Case 3) was more effective in minimizing the severe discomfort hours than rendering the internal walls and the inner part of external walls (Case 9). Previous studies reported that increased insulation results in overheating in buildings [61,62], which is somewhat consistent with the current study's findings that incorrect insulation application may lead to higher discomfort hours. In the latter case (Case 9), the heat transfer rate between bedroom 4 and comparatively cooler adjacent family zone decreases due to the application of aerogel render on internal walls. The family zone is comparatively cooler because it has a shaded north window and other zones act as a buffer on three sides.

Furthermore, application of PCM and aerogel render on the outer side of the external walls (Case 6), on internal walls and the interior side of the external walls (Case 12), and the insulated ceiling (Case 7, Case 13) further reduced the severe discomfort hours compared to insulated ceiling case (Case 3 and Case 9). Although the retrofit Case 6 was more effective in minimizing discomfort hours compared to Case 12, the pattern changed with the integration of PCM in ceilings. Figure 8 shows that the integration of PCM and aerogel render on the inner part of the external wall, internal walls, and insulated ceiling (Case 13)

was the best strategy to reduce severe discomfort hours. It could be because of the large applied surface area and a thinner layer of PCM that accelerated solidification and melting in Case 13 and absorbed any trapped heat.

The optimum PCM temperature (OPCT) was calculated based on the maximum reduction of severe discomfort hours (SDH). Figure 8 shows that OPCT depends on the application method of PCM. The OPCT was mainly in the range of 22–25 °C when PCMs were applied on the inner and outer parts of the wall. In Case 4, the OPCT was found to be in the range of 24–32 °C, which means there was no change in discomfort hours when PCMs in this temperature range were applied in Case 4. Moreover, in the case of PCM and aerogel render on the outer part of the external wall and in the uninsulated ceiling (Case 4), the OPCT was in the range of 29–32 °C. For Case 7, severe discomfort hours were minimum when phase temperature was within 29–32 °C. No conclusive evidence was found on the impact of PCM position (inner and outer parts of the wall) on OPCT.

### 3.2. Performance of Retrofitting Strategies in Terms of Energy Savings

Figure 9 shows annual heating and cooling energy demand in different simulation cases. The calculated heating loads are generally much higher than the cooling load because Melbourne is in a cool temperate climate zone.

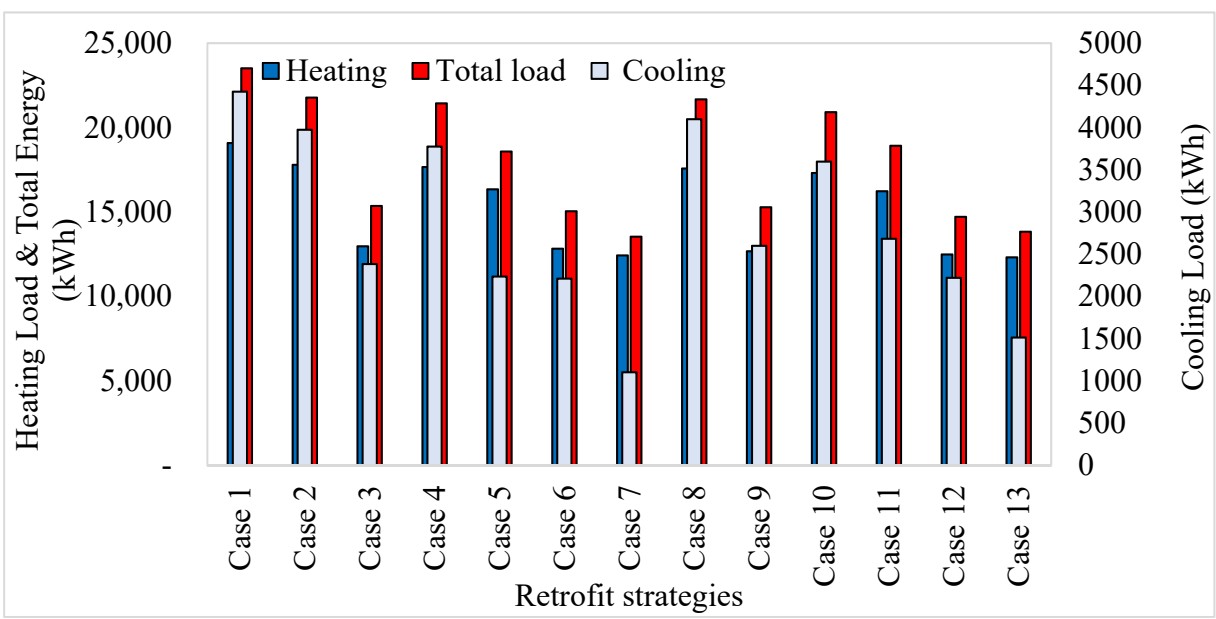

**Figure 9.** Annual cooling, heating, and total loads in reference and retrofitted buildings.

The figure shows that both heating and cooling energy load decreased significantly compared to the reference case (Case 1) with ceiling insulation and aerogel render (Cases 3 and 9). However, aerogel rendering the outer part of the external wall (Case 3) was found to reduce the cooling load slightly higher (47%) than the interior aerogel rendering (41%) (Case 9). On the other hand, the heating load reduction was marginally higher in interior rendering (34%) than exterior rendering (32%). This is in line with the observation in Section 1, where it was reported that rendering the outer side of the external wall minimizes the overheating effect more than the interior rendering. In Cases 5 and 11, the application of PCM in the non-insulated ceiling and aerogel rendered walls significantly reduced the cooling loads (50–53%), which was higher than that of Cases 3 and 9. The reduction in heating loads in Cases 4 and 11 (15–17%) was much lower than that of Cases 3 and 9. However, the total energy consumption in Cases 3 and 9 reduced by 35%, which is much higher than Case 5 and Case 11 (22%), hence they are more preferred options.

Moreover, integration of PCM in insulated ceilings and aerogel rendered walls (Cases 7 and 13) resulted in further significant reduction in cooling load compared to Cases 3 and 9,

but the reduction in heating load was marginal. Overall, in Cases 7 and 13, total energy, cooling energy, and heating energy consumption reduced by 41–42%, 69–70%, and 34–36%, respectively, compared to Case 1. Case 13 showed marginally higher heating energy savings but lower cooling energy savings than Case 7. Therefore, in terms of total energy savings, Case 13 is a slightly better option than Case 7. However, as mentioned above, it may be more practical to select Case 7 to avoid interruption in daily activities during retrofitting. The integration of PCM in an insulated ceiling was more efficient than applying PCM only. Therefore, in terms of retrofitting, insulation of the ceiling should come first, and PCM and aerogel render application further reduce the energy use by increasing the heat storage capacity of the building envelope.

Overall, the cooling energy savings potential of different retrofit strategies is much higher than heating energy savings because the PCM layer resists and absorbs heat transfer through the envelope by phase transition (liquefaction and solidification) during summer. In winter, the phase transition activities are very limited due to unfavorable outdoor weather, and the PCM layer mainly resists heat transfer through the envelope being in a solid-state. A different phase transition temperature may be required to enhance the heating energy-saving potential. To further understand this matter, the optimum phase change temperatures in terms of heating, cooling, and total energy savings are presented in Figure 10.

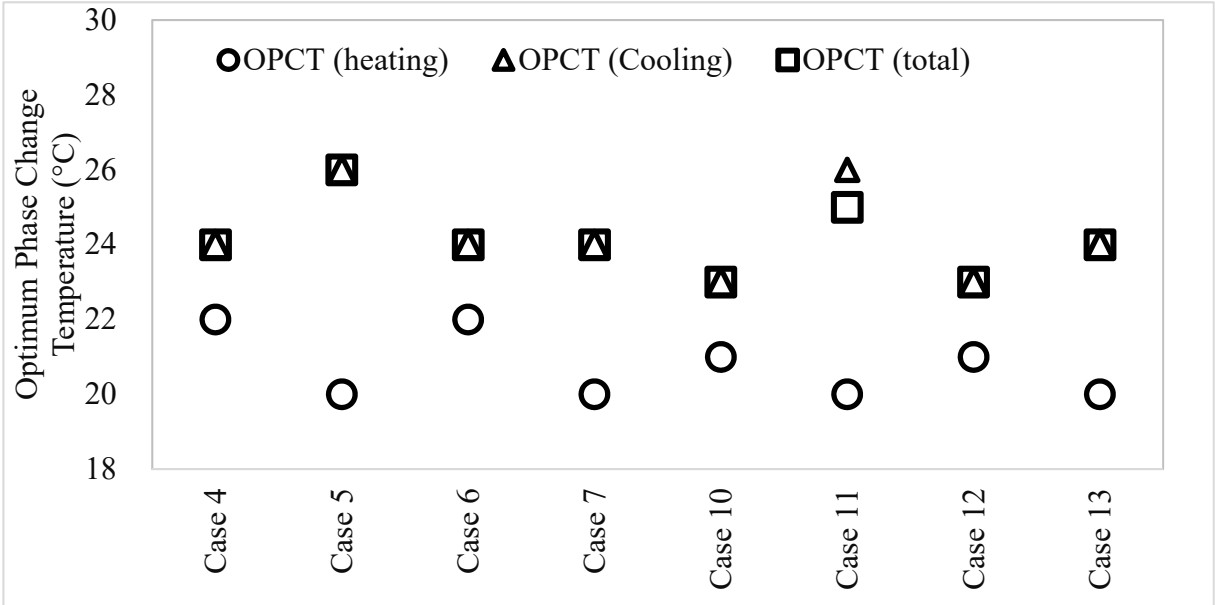

**Figure 10.** Optimum phase change temperatures for heating, cooling, and total energy savings.

Figure 10 shows that the optimum phase change temperature (OPCT) for heating is around 20–22 °C, which is close to the HVAC heating set-point 20 °C of the building. Similarly, the OPCT for cooling is found to be around 23–26 °C, which is close to the HVAC cooling set-point of 24 °C. However, it is not practical to integrate two different phase change materials with optimum temperature for heating and cooling, thereby OPCT is selected based on annual energy use. Figure 10 also shows the OPCT for total energy consumption, which is mostly similar to the OPCT for cooling. Figure 11 shows total annual energy savings for PCM-integrated cases with three different OPCT: OPCT heating, OPCT cooling, and OPCT total. Although OPCT for heating can maximize the heating energy savings, their total energy-saving performance is lower than the OPCT cooling and OPCT total cases. Therefore, OPCT corresponding to cooling or total energy savings is preferred in these cases.

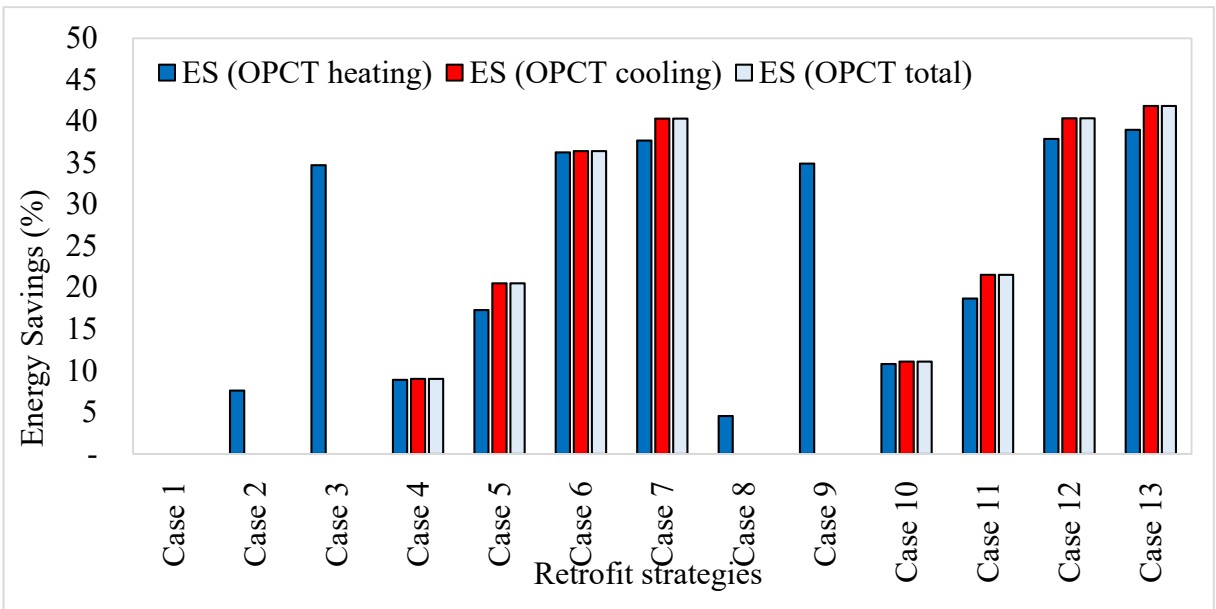

**Figure 11.** Annual energy saving corresponding to optimum phase change temperatures for heating, cooling, and total energy.

Finally, the OPCT corresponding to cooling or total energy savings (23–26 °C) is much lower than the OPCT corresponding to minimum heat stress risk (30–32 °C) for all cases reported in Section 1. Therefore, the selection of OPCT depends on whether a building is free running or mechanically cooled.

### 3.3. Peak Cooling Load Reduction

Figure 12 shows hourly peak cooling loads of different retrofit scenarios. As expected, the peak cooling load in the reference case without any retrofit measures (Case 1) is the highest compared to other cases in the living room of the studied house. However, the figure also shows that the integration of ceiling insulation is crucial to reduce the peak cooling load. In the absence of ceiling insulation, PCM combined with aerogel render (Cases 2, 4, 5, 8, 10, and 11) decreased the peak cooling loads between 5–24%. The integration of ceiling insulation with other retrofit measures significantly reduced the peak cooling load.

Case 3 and Case 9 reduced the peak cooling load by 47% and 45%, respectively, which was further reduced with the application of PCM in the ceiling and walls. The best retrofit scenario for minimizing the peak cooling load was Case 7, with a 65% reduction in peak cooling load.

### 3.4. Performance of Retrofitting Strategies in Terms of Operational Emission

The emission resulting from natural gas consumption and electricity use is illustrated in Figure 13. The emissions of different retrofit strategies with PCM were calculated using OPCT corresponding to the cooling load mentioned in Section 1. Although the building has a comparatively lower cooling load than heating (as shown in Figure 9), the greenhouse gas (GHG) emissions resulting from cooling are higher than emissions associated with heating. Hence, the unit heat produced by natural gas has a lower emission factor than unit electricity used for cooling. Among the retrofitting strategies, aerogel rendered walls (Cases 2 and 8) and aerogel-based render coupled with PCM wall (Cases 4 and 10) have little impact on heating and cooling load emission reduction. However, the application of PCM in the walls and ceiling (Case 5 and Case 11) halved the cooling load emission with a slight decrease in heating load emission, which was further reduced by insulating the ceiling (Cases 6 and 12). The cases with PCM in the insulated ceiling and rendered walls

(Cases 7 and 13) resulted in a maximum 64% total reduction in GHG emission than the base case (Case 1). Out of that 64%, 70% is due to the cooling load emission reduction. Therefore, the PCM application is most beneficial because a significant portion of GHG emissions is associated with the cooling load.

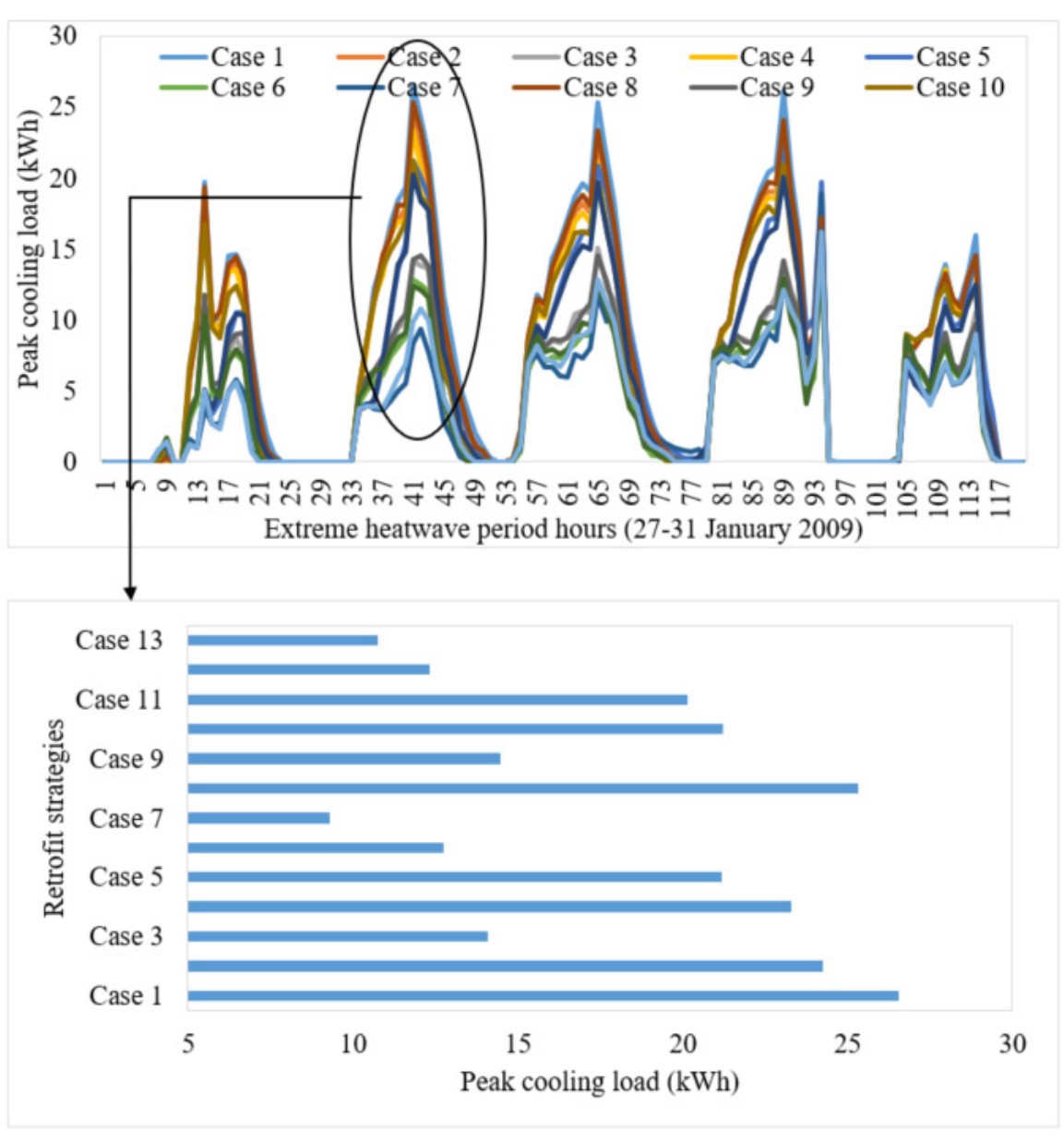

**Figure 12.** Peak cooling load of building under different retrofit strategies.



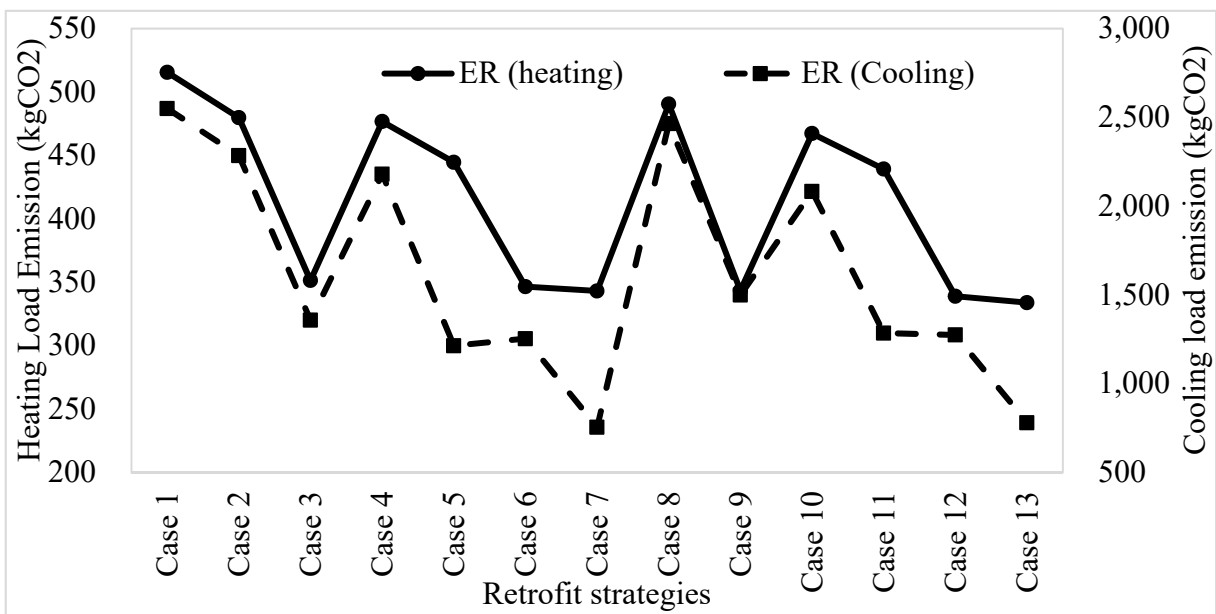

**Figure 13.** $CO_2$ emission associated with natural gas consummation and electricity use in the retrofitted building.

*3.5. Performance of the Retrofitting Strategies in Terms of Cost*

The operational energy cost and initial investment cost of different retrofit strategies are presented in Figure 14, and their respective cost savings are shown in Figure 15. As seen in Figure 14, the operational energy cost (sum of unit electricity cost and natural gas) was reduced by the adding aerogel render, PCM, and insulation. In this study, PCM and insulation quantity were kept constant in all cases because PCM performance varies with its heat storage capacity, which depends on its quantity, while insulation is only applied to the ceiling. The OPCT corresponding to maximum total energy savings was used to evaluate the economic performance of proposed retrofit strategies. The figure shows that applying PCM on the rendered wall and the uninsulated ceiling has an insignificant impact on operation energy cost. However, insulating the ceiling dramatically reduced the operation cost because heat transfer through the roof was higher than the wall [60]. As a consequence, the operation cost dropped to $3.93–3.96 k/year excluding PCM (Cases 3 and 9), which was further reduced to $3.83–3.89 k/year and $3.65–3.73 k/year by applying PCM on the walls (Cases 6–10), and walls and ceiling (Cases 7 and 13), respectively. Although Case 13 was found to have the lowest operational energy cost, the initial investment, in this case, was significantly higher ($30k) compared to Case 7 ($18k). Therefore, Case 7 may be preferred from the perspective of lower investment cost and quicker payback period.

Figure 15 shows that the retrofit strategies with insulated ceilings (Case 3, 6, 7, 9, 12, and 13) resulted in positive lifecycle cost savings and can be considered cost-effective. Among all strategies, Case 3 was found to have the highest cost savings over the 40 years lifetime period. The life cycle cost-saving decreases with the integration of PCM and rendering compared to the insulation-only case (Case 3) because of the higher investment cost associated with PCM and render. With aerogel render, PCM, and ceiling insulation, Case 7 resulted in the highest lifecycle cost savings.

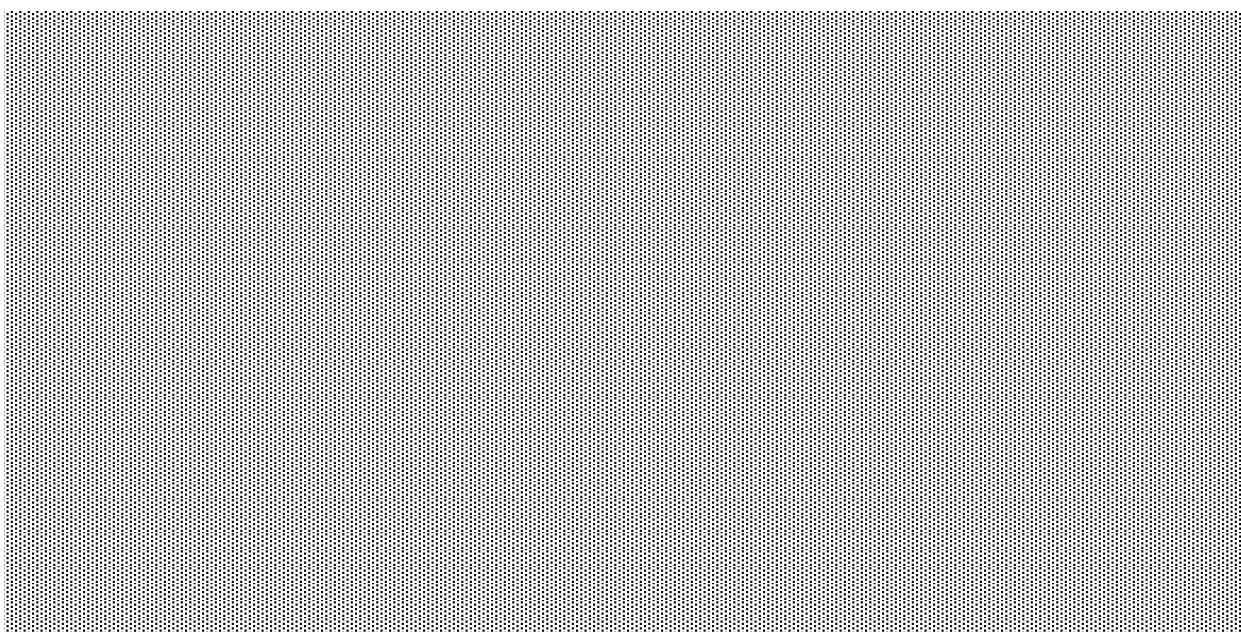

**Figure 14.** Initial investment and annual operation energy cost in different retrofit cases.

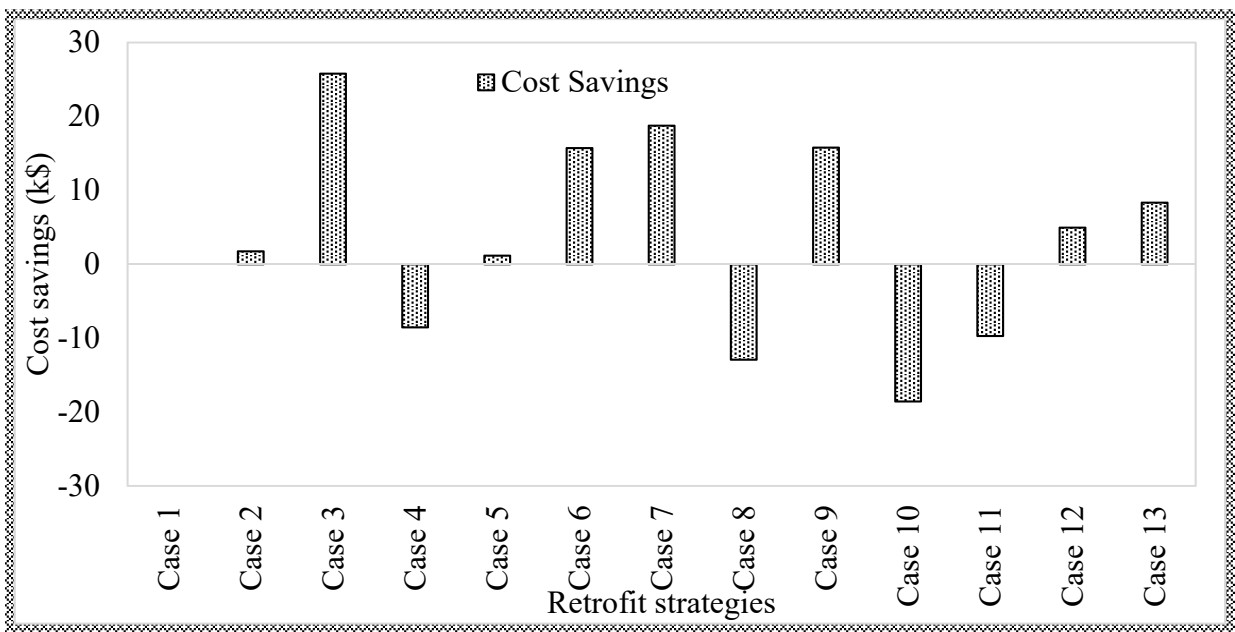

**Figure 15.** Lifecycle cost savings of different retrofit strategies.

## 4. Discussion

The selection of the best retrofit strategy depends on the will of building stakeholders. The private stakeholders are more concerned about thermal comfort and cost savings, while public stakeholders stress on energy-efficient and eco-friendly building design. That is why the application of aerogel renders, PCM, and insulation was evaluated considering the improvement in thermal comfort, increased energy savings, reduction in emission, and maximum lifecycle cost savings. This study assessed 12 different retrofit strategies for mitigating overheating risk in a non-air-conditioned house and minimizing the peak cooling demand, annual energy use, emission, and cost savings in an air-conditioned residential building.

The comparative analysis of results revealed that Case 13 was the best retrofit option to minimize the severe discomfort hours, total heating and cooling energy consumption,

and annual operational cost. On the other hand, Case 7 was found to be the best option to minimize peak cooling load during a hot summer period and total $CO_2$ emission associated with heating and cooling load. Also, total lifecycle cost savings were higher for Case 7 than Case 13 because the retrofit investment cost was much higher in the latter case. Therefore, Case 7 may be preferred over Case 13, considering significant lifecycle cost savings, although the performance of Case 7 is marginally lower in terms of discomfort hours, total energy, and operational cost. Furthermore, Case 7 may also be preferred to avoid interrupting occupants' daily life, which is one of the key building energy retrofitting barriers [63]. Finally, the peak cooling load performance of Case 7 was the best among all studied options. As mentioned in 3.3, reducing peak cooling load is very important to eliminate power outages and reduce electricity infrastructure costs that will otherwise be required to meet the peak demand. Hence, Case 7 can be considered as the best retrofit option with PCM.

However, lifecycle cost savings of Case 3 were found to be highest amongst all simulated cases. A comparison between Case 3 and Case 7 showed that lifecycle cost savings of Case 3 is 27% higher than Case 7. However, for the latter case, the severe discomfort hours, total energy consumption, peak cooling load, $CO_2$ emission, and annual operating cost are 64%, 9%, 14%, 36%, and 6% lower than Case 3. Hence, Case 3 can be considered if the cost is the primary selection criteria, as in the case of private stakeholders mentioned above. However, Case 7 could be preferred by public stakeholders with more emphasis on energy-efficient and eco-friendly building design.

### 4.1. Impact of Phase Change Temperature on Performance Indicators

Once Case 7 was selected as the best retrofit strategy, the next key task was selecting the optimum PCM temperature for Case 7, considering the comfort hours, energy savings, peak demand, cost savings, and emissions. Figure 16 shows the performances of Case 7 with different PCM temperatures. The figure shows that the optimum phase change temperature for the minimum severe discomfort hours lies between 29 to 32 °C (also discussed in Section 1). On the other hand, 25 °C PCM results in maximum annual energy savings (40%), emission reduction (63.58%), and lifecycle cost savings ($18.75 k). Furthermore, the peak cooling load was the lowest (9.3 kW) with PCM between 24 and 26 °C. Therefore, if the primary aim is to reduce the thermal discomfort hours in a naturally ventilated (free-running) house during a heatwave period, 29–32 °C should be preferred. However, in an air-conditioned or mixed-mode building, 25 °C PCM is recommended.

However, it should also be noted that although 29–32 °C PCM results in a maximum reduction in discomfort hours during a severe heatwave period, it may not be suitable to increase thermal comfort during the rest of the years. Because approximately 69% of Australians use air conditioner during the hot summer period, 25 °C PCM would be ideal. If there is a power outage during a heatwave, the selection of 25 °C PCM only increases the severe discomfort hours by 3 h during a heatwave period (from 5 to 8 h), but is still 71% lower than the base case scenario mentioned in Figure 8. On the other hand, the use of 29 °C instead of 25 °C in an air-conditioned house increases the peak cooling demand by 6% and decreases the annual energy savings, emission reduction, and cost savings by 2%, 18%, and 28%, respectively. These changes are far more significant than the changes in severe discomfort hours. Therefore, 25 °C PCM can be considered as the optimum PCM for Case 7.

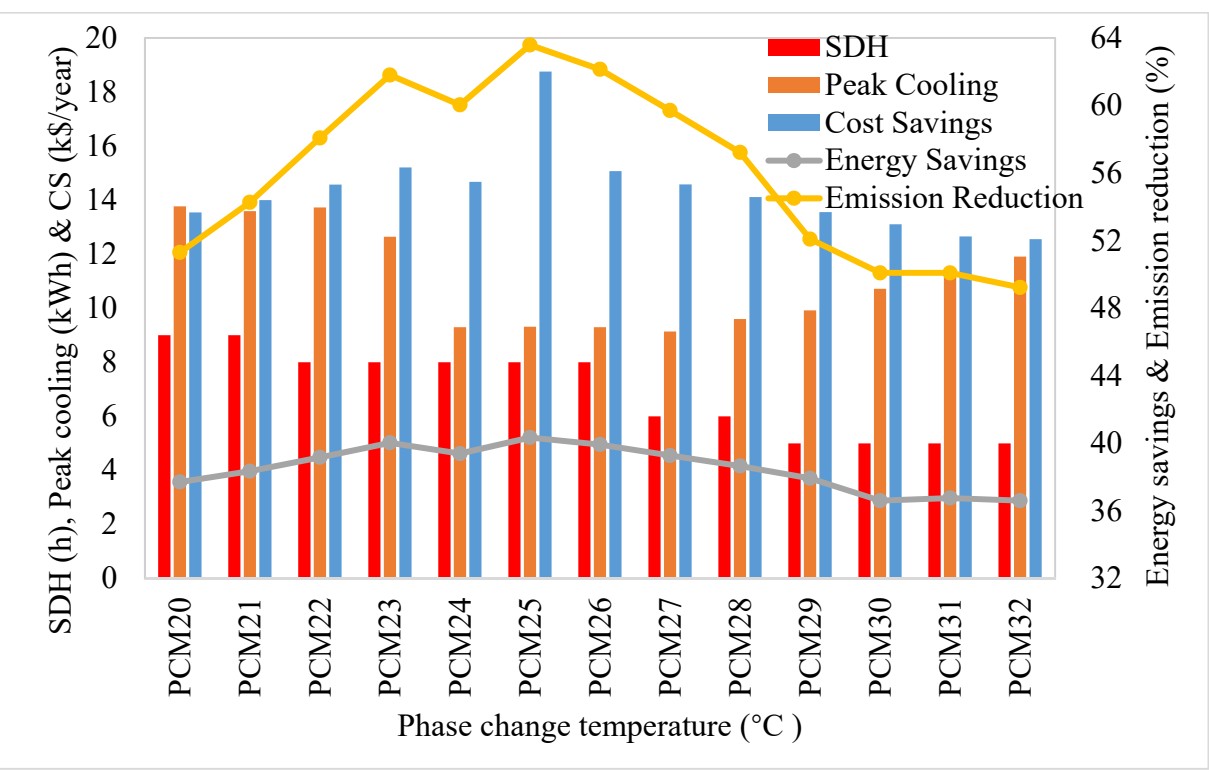

**Figure 16.** Impact of phase change temperature on severe discomfort hours (SDH), peak cooling demand, energy savings, emission reductions, and cost savings.

### 4.2. Impact of PCM and Aerogel Render Thickness on Performance Indicators

Figure 17 shows that severe discomfort hours decrease with increasing render thickness at varying degrees depending on the PCM thickness. Minimum severe discomfort hours can be achieved either by having a combination of thicker render and thinner PCM or with a thinner render and thicker PCM as shown in Figure 17. A thicker render increases resistance to heat transfer from ambient air, and a thicker PCM can absorb more heat from the ambient air. Figure 17 shows that at 0.02 m render thickness, a PCM layer of 0.0225 m is required to achieve minimum discomfort hours. On the other hand, for 0.05 m render thickness, a 0.005 m PCM layer is sufficient to achieve the minimum discomfort hours.

Figure 18 exhibits the influence of aerogel render thickness on annual energy use in an air-conditioned building at different PCM thicknesses. Annual energy use decreases with an increase in both aerogel render and PCM thickness. However, the impact of increasing aerogel render thickness is higher than PCM panel thickness because aerogel render has higher thermal resistance than PCM. Moreover, the building is located in a heating-dominated region where it is desired to retain heat within an occupied space without transferring much into the ambient environment. For instance, the annual energy use was reduced by 550 kWh, with increasing render thickness from 0.01 m to 0.05 m at constant PCM thickness (0.005 m). On the other hand, increasing PCM thickness from 0.005 m to 0.025 m at a constant render thickness of 0.01 m only reduced the energy use by 200 kWh. The thickest layer of PCM (0.025 m) and aerogel render (0.05 m) on the outer part of the wall resulted in the lowest annual energy use because of being heat resistive; aerogel render saves heating energy, whilst, being heat-storage materials, PCM saves cooling energy use. However, this combination also results in the highest investment cost, and therefore, the optimum combination needs to be selected considering the other performance criteria.

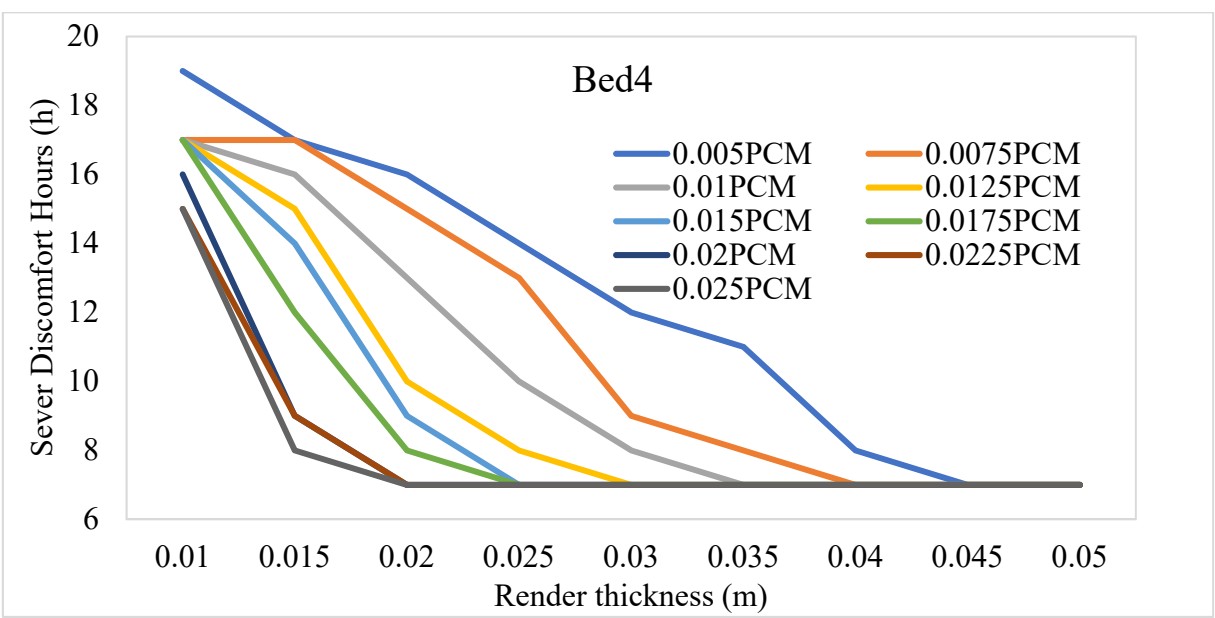

**Figure 17.** Impact of aerogel render thickness on sever discomfort hours considering the different thicknesses of PCM on the exterior side of the wall.

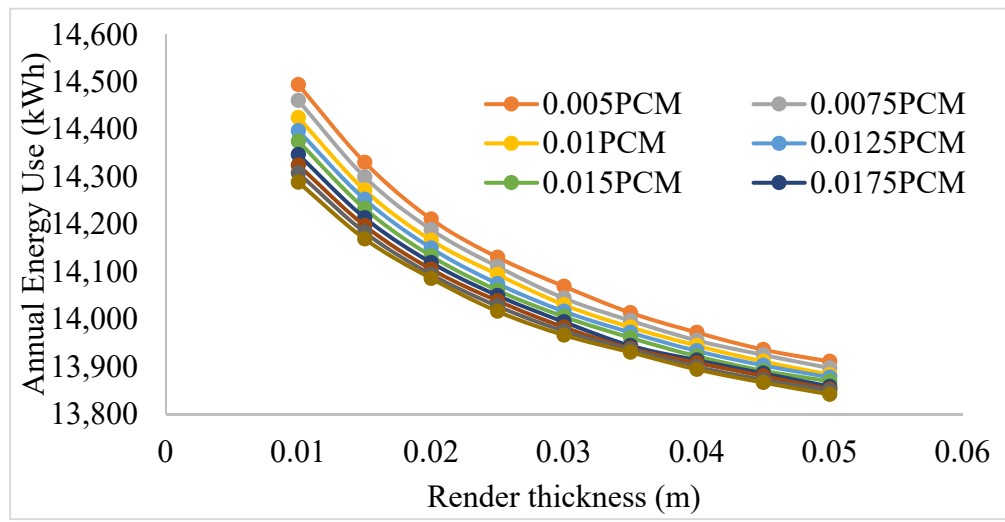

**Figure 18.** Impact of aerogel render thickness on annual energy consumption considering the different thicknesses of PCM on the exterior side of the wall.

Increasing render and PCM thickness also reduced $CO_2$ emission associated with heating and cooling energy use, as shown in Figure 19. Compared to energy, the degree of emission reduction with increasing PCM thickness is comparatively higher at a constant aerogel render thickness. This was because PCM helps to reduce cooling energy demand and the emissions associated with cooling energy use are higher than heating due to their emission factors as discussed in Section 1. The lowest $CO_2$-emission was found for a 0.05 m-thick rendered wall and 0.025 m PCM. An increase in PCM thickness at higher render thickness meagerly reduced $CO_2$ emission compared to lower render thickness.

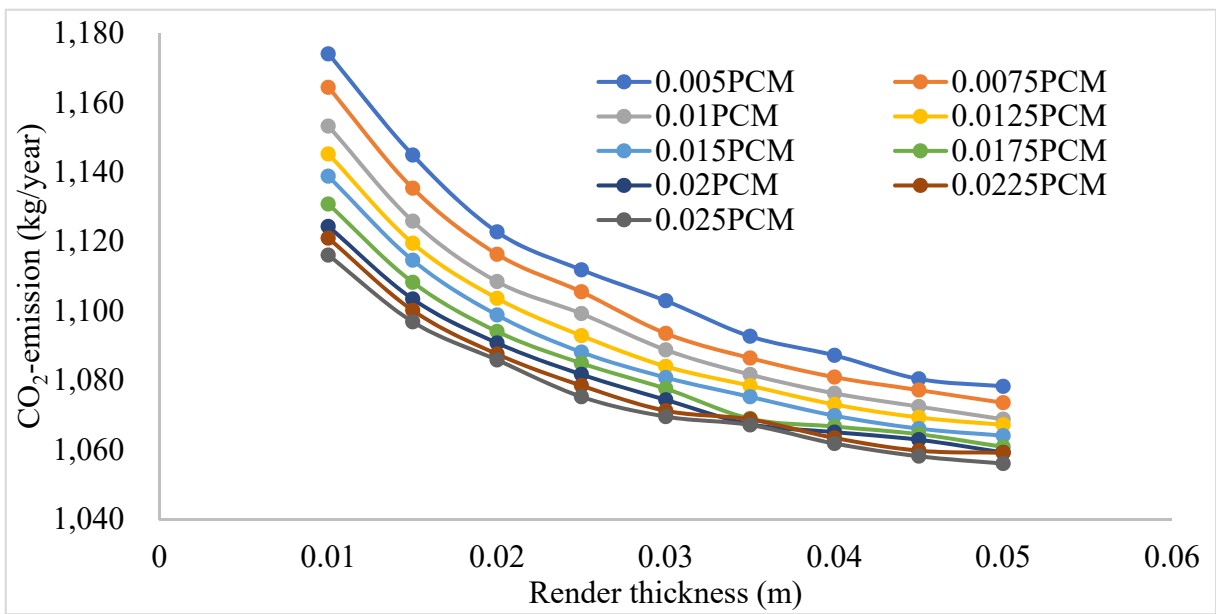

**Figure 19.** Impact of aerogel render thickness on emission considering the different thicknesses of PCM on the exterior side of the wall.

Figure 20 shows variations in life cycle cost savings with different aerogel render and PCM thickness. All combinations of PCM and aerogel render thickness are economically feasible due to positive lifecycle cost savings. However, the savings decrease significantly with the increasing thickness of aerogel render and PCM due to higher initial investment cost. Figure 20 also shows that the cost savings decrease sharply with increasing aerogel render thickness compared to increasing PCM thickness, because the cost of aerogel render is 10 times of PCM. From an economic perspective, a 0.01-m-thick aerogel render and 0.005-m-thick PCM layer should be applied on the outer side of the wall for the highest cost savings among different combinations of PCM and aerogel render thicknesses. However, this combination results in maximum annual energy use and emission, as shown in Figures 18 and 19. Hence, there is a need to find the optimum thickness considering costs, energy, and emission.

Figure 21 shows two Pareto optimization curves created using lifecycle cost, energy consumption, and emission. Pareto font consists of a non-dominated solution where there is no other feasible solution to improve one objective without deteriorating others. The optimum single solution that satisfies the multiple objectives would be selected based on the utopia point criterion. Here, the utopia point represents the point with the lowest lifecycle cost and energy consumption (Figure 21a) and life cycle emission (Figure 21b). The solution is close to the utopia point, which was considered the optimum PCM and aerogel render thickness. The optimum thickness was found to be 0.025 m for both PCM and aerogel render based on both energy consumption and emission. This is different from the best thickness combination identified based on the energy and emission earlier. The optimum solution is highlighted in red in Figure 21 with $84,855 lifecycle cost, 2018 GJ lifecycle energy consumption, and 43 tons of $CO_2$-e emission. This thickness combination results in $24,000 lifecycle cost savings as seen from Figure 21. The identified optimum aerogel-render (0.025 m) and PCM (0.025 m) thickness combination is also suitable to achieve minimum discomfort hours.

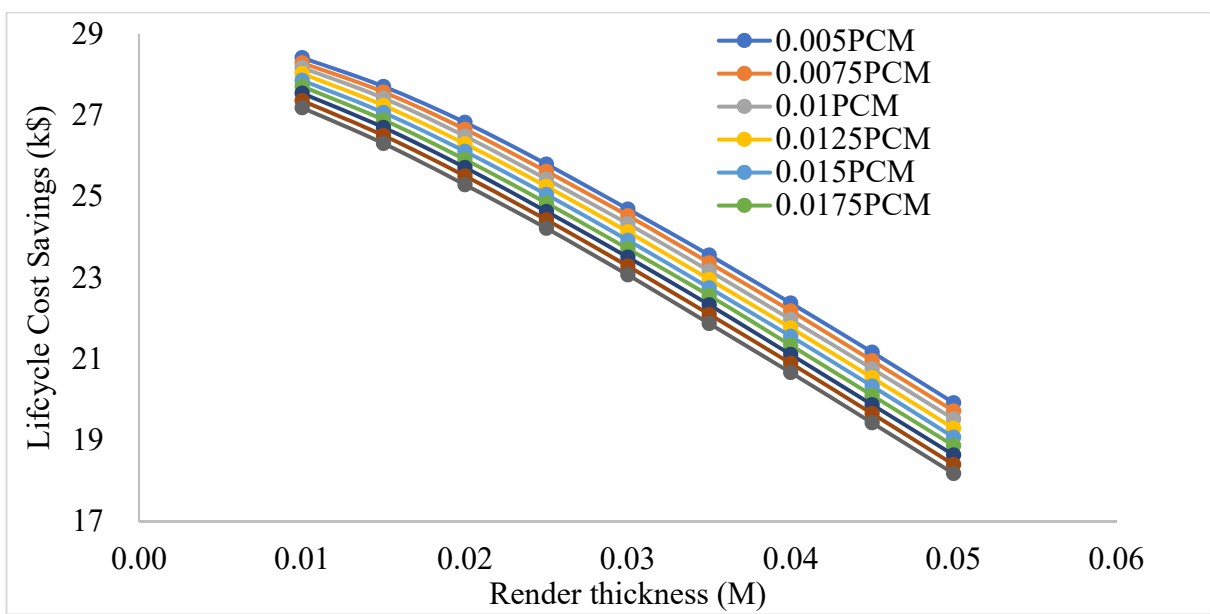

**Figure 20.** Lifecycle cost savings for different aerogel render and PCM thickness on the exterior side of the wall.

This study did not consider supercooling and hysteresis effect of BioPCM due to the lack of available data. Therefore, it may have resulted in some inaccuracies in the energy-saving performance calculation of PCM. PCM with high supercooling may not solidify entirely at night and result in lower cooling energy-saving potential [19]. However, organic PCM generally has a shallow supercooling effect. PCM with high hysteresis improves the thermal performance of PCM walls. Paraffin has low hysteresis with a more negligible difference in melting and solidification curve within 1.2 °C [64].

Moreover, PCM-hysteresis resulted in the mean relative error in the simulated wall's surface temperature and heat flux of 3.5 and 5% compared to PCM without hysteresis [65]. Therefore, the exclusion of PCM-hysteresis may impact heating and cooling energy consumption; however, the impact will be uniform for all simulated cases. Hence, this will not change the critical findings of this study regarding optimum retrofit combinations and optimum PCM temperature, PCM, and aerogel thickness.

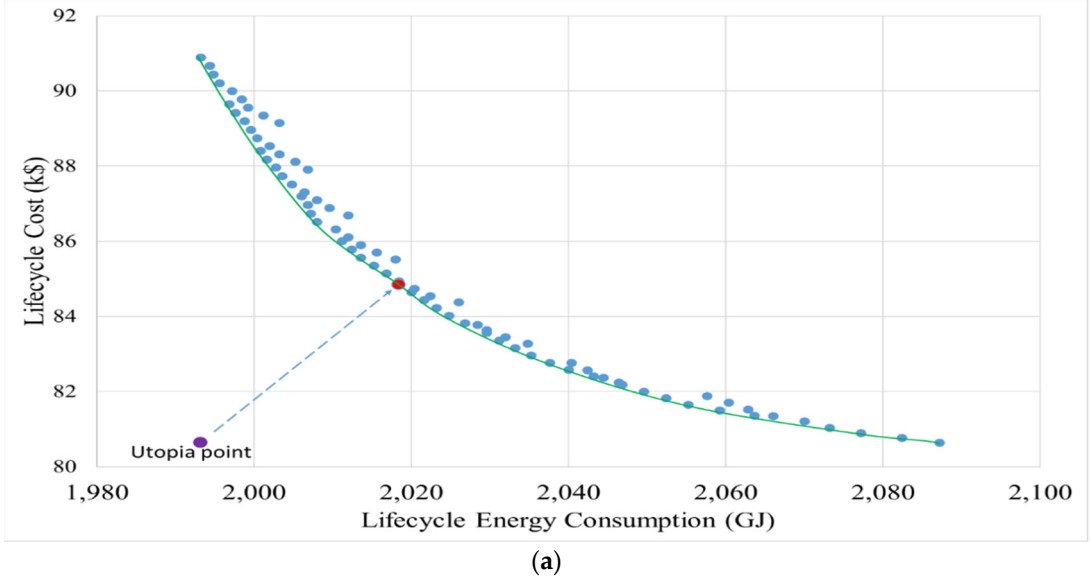

(**a**)

**Figure 21.** *Cont.*

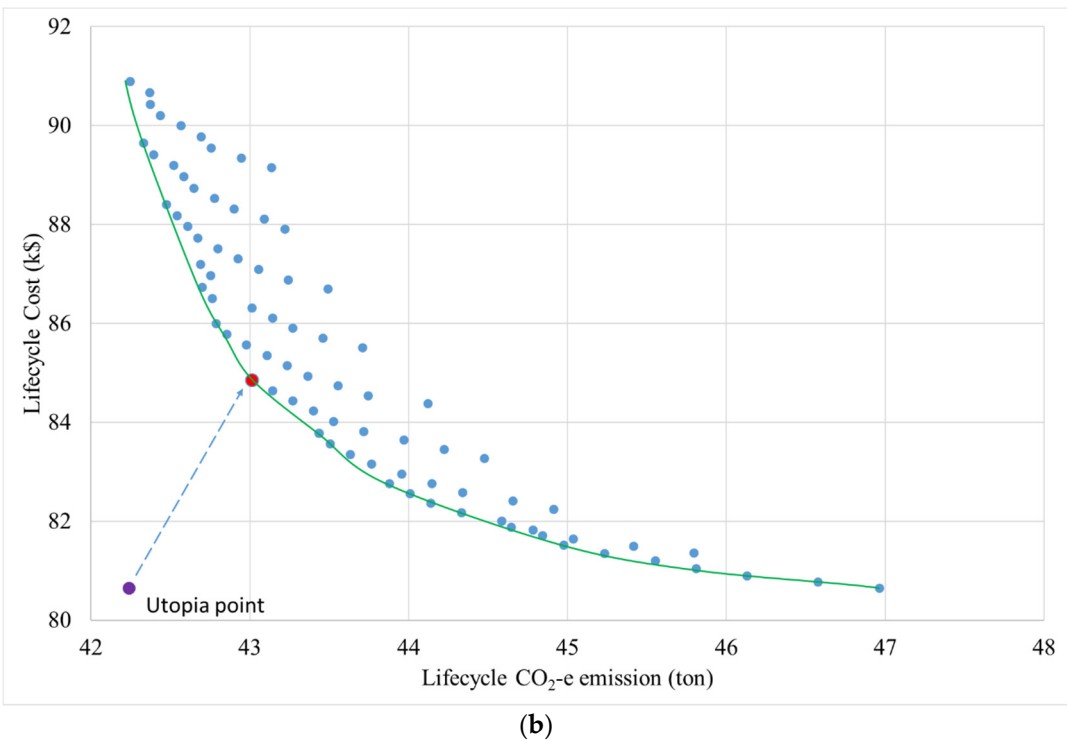

(**b**)

**Figure 21.** Pareto multi-objective optimization curve for (**a**) lifecycle cost vs lifecycle energy consumption, and (**b**) lifecycle cost vs lifecycle $CO_2$-e emission.

## 5. Conclusions

This study numerically investigated 12 different building envelope retrofit strategies, including aerogel render, PCM, and insulation using the building simulation tool EnergyPlus v9.2. The performance of proposed retrofit strategies was evaluated considering overheating risk, energy efficiency, peak cooling load, emission reduction, and cost savings.

The aerogel-based render coupled with PCM outside of external walls and PCM combined with insulated ceilings (Case 7) was found to be the best retrofit strategy considering all performance categories. Compared to the baseline, this strategy reduced severe discomfort hours, total energy consumption, peak cooling load, $CO_2$ emission, and operational energy cost by 82%, 40%, 65%, 64%, and 35%, respectively. Although the lifecycle cost savings of Case 7 are lower than Case 3 (insulated ceiling and externally rendered wall) because of the high investment cost of PCM, the former one can be selected considering its higher environmental performance. Mainly, this would be preferred by public stakeholders where the stress is on energy-efficient and eco-friendly building design. The 25 °C melting point PCM was considered the best option to minimize severe discomfort hours (in non-air-conditioned houses and during the blackout period) during a heatwave period as well as to reduce total energy, emission, cost, and peak cooling load. Parametric analyses showed that the thicker the PCM and aerogel render, the lower is the energy consumption and emission. However, increased PCM and aerogel render thickness decreased lifecycle cost savings due to high investment costs. The optimum thickness for PCM and aerogel render was 0.025 m considering the emission, comfort, energy, and life cycle costs for a typical Australian house in Melbourne climate. This strategy (Case 7) will have a minimum interruption to occupants' daily life while retrofitting because of being applied outside of the external wall.

This study is the first step of a PCM-integrated aerogel render development project. In the future, the findings from this simulation study will be used to develop a PCM-integrated aerogel render. The thermal properties and performance of the developed render will be evaluated experimentally. Then the numerical model developed in this study will be updated to include the properties of PCM-integrated aerogel render, and its

performance will be compared against the results presented in this study where PCM and aerogel renders were assumed as separate layers. Furthermore, the thermal performance of the PCM-integrated aerogel render will be evaluated for different climate zones.

**Author Contributions:** Conceptualization, D.K., M.A. and J.G.S.; methodology, D.K.; software, M.A.; validation, M.A.; formal analysis, D.K.; writing—original draft preparation, D.K.; writing—review and editing, M.A.; supervision, M.A, J.G.S.; project administration, J.G.S.; funding acquisition, D.K. All authors have read and agreed to the published version of the manuscript.

**Funding:** This research was funded by Higher Education Commission of Pakistan grant number No. 5-1/HRD/UESTPI(Batch-VI)/6021/2018/HEC.

**Institutional Review Board Statement:** Not Applicable.

**Informed Consent Statement:** Not Applicable.

**Data Availability Statement:** Not applicable.

**Acknowledgments:** Authors would like to thanks the Higher Education Commission (HEC), Pakistan for supporting their research work under the HEC-SUT joint scholarship.

**Conflicts of Interest:** The authors declare no conflict of interest.

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
