# Peer review of "Retrofitting Building Envelope Using Phase Change Materials and Aerogel Render for Adaptation to Extreme Heatwave: A Multi-Objective Analysis Considering Heat Stress, Energy, Environment, and Cost"

_sustainability, doi:10.3390/su131910716_

Round 1

Reviewer 1 Report

The manuscript deals with the assessment of PCM & aerogel render suitability for energy retrofit of a typical Australian family house in extreme climate. A total of 13 cases (baseline case + 12 retrofit cases) for two different strategies (with and without HVAC system) were evaluated and compared in terms of: Indoor heat stress and thermal discomfort, energy savings, emission reduction, life cycle cost analysis. For the "best" retrofit combinations, PCM and aerogel render thickness were also analysed.
The topic is highly relevant as changes in future climate will lead to increased cooling demand in buildings worldwide and corresponding retrofit solutions with innovative and advanced materials will be necessary.
However, in my opinion, the paper lacks clarity and some drawbacks must be improved and/or clarified.

#1 Introduction section: in the part where you define the aims of your study (p. 8/55), when referring to PCM and aerogel render you use the terms "incorporate" and "integrate". To my knowledge, this implies that PCM and aerogel are "mixed" into one product (encapsulating PCM in aerogel render). However, in your simulations you modelled PCM and aerogel render separately, i.e. as different layers. My first impression was that you modelled PCM blanket or square pouch covered by aerogel render.
Can you please clarify this? How did you envision your combination of PCM and aerogel render? If PCM is incorporated in aerogel render, do you think it is acceptable to model them as separate layers?

#2 Table 2: Thermophysical properties of aerogel render are missing.

#3 Is there any particular reason why you chose the Melbourne climate?

#4 Since you did not perform the EnergyPlus validation as part of your study, it should be moved from section 2.2 Building Energy and Thermal Simulations to 1. Introduction section (or perhaps reorganise the Introduction section and add an additional Literature Review section).

# 5 Table 4: It is not clear from the description what you mean by "Applying render on the outer/inner parts of the wall". Which walls? External walls? Internal walls?
Reader must carefully compare Table 4 with Table 5 to get the whole picture. Perhaps you could combine these two tables into a single table.

# 6 Table 5 - ceiling insulation? I do not see its properties in Table 2. What is the construction detail of the ceiling? Is it a concrete slab or a wooden frame ceiling? Not clear... If it is a wooden frame ceiling, of course there is insulation between the wood rafters. That should be baseline case, not the retrofit case.
It is also not clear what the construction detail of external wall is. From Table 5, one might conclude that External wall is just brick veneer and plasterboard. This type of construction usually also consists of an air cavity and insulation, as you have model it in another publication https://www.sciencedirect.com/science/article/abs/pii/S0378778814003405

#7 Please explain why you placed your PCM on the outer side of external wall (Cases 4 - 7)? Isn't the main point to locate PCM on the side of the conditioned zone so that it can respond to temperature changes of the indoor air rather than the external air? In Cases 4 - 7, placing PCM on the outer side of external wall it is exposed to the temperature changes of the outside air.
In Cases 10-13, on the other hand, you have placed PCM on the inner side of external wall (on the side of the conditioned zone). Now, cases e.g. 4 and 10 cannot be compared to study the effect of retrofitting the internal walls as well.

#8 Section 2.5.1 Indoor heat stress risk and thermal discomfort: how did you determine the wet bulb temperature?

#9 Is there a specific reason why your diagrams are not visually consistent throughout the manuscript? For example, compare Figure 12 or Figure 16 with Figures 8, 9, 11, etc.

This research is promising, but certain aspects need to be critically checked, well argued, and thoroughly revised. 

Author Response

Comment 1.1 : The manuscript deals with the assessment of PCM & aerogel render suitability for energy retrofit of a typical Australian family house in extreme climate. A total of 13 cases (baseline case + 12 retrofit cases) for two different strategies (with and without HVAC system) were evaluated and compared in terms of: Indoor heat stress and thermal discomfort, energy savings, emission reduction, life cycle cost analysis. For the "best" retrofit combinations, PCM and aerogel render thickness were also analysed.

The topic is highly relevant as changes in future climate will lead to increased cooling demand in buildings worldwide and corresponding retrofit solutions with innovative and advanced materials will be necessary.
However, in my opinion, the paper lacks clarity and some drawbacks must be improved and/or clarified.

Response 1.1 : We thank the reviewer for the constructive comments which certainly improved the quality of our manuscript.

Comment 1.2: Introduction section: in the part where you define the aims of your study (p. 8/55), when referring to PCM and aerogel render you use the terms "incorporate" and "integrate". To my knowledge, this implies that PCM and aerogel are "mixed" into one product (encapsulating PCM in aerogel render). However, in your simulations you modelled PCM and aerogel render separately, i.e. as different layers. My first impression was that you modelled PCM blanket or square pouch covered by aerogel render.
Can you please clarify this? How did you envision your combination of PCM and aerogel render? If PCM is incorporated in aerogel render, do you think it is acceptable to model them as separate layers?

Response 1.2:   Great question. Our goal in this project is to develop a PCM integrated aerogel render for easy retrofitting of existing building wall. This simulation study was carried out as part of the feasibility study to know how the combination of PCM and aerogel influences building thermal performance and energy consumption. While we know the thermal properties of PCM and aerogel separately, we don’t know yet the thermal properties of PCM integrated aerogel render. Therefore, for the purpose of this feasibility study and for the sake of simplicity, we have assumed PCM and aerogel render as a separate layer in this simulation study. This study provides important information regarding the required PCM properties and thermal resistance of the aerogel PCM layer which will be used to develop the PCM integrated aerogel render. Once we develop the render, we will measure the thermal properties of that render and conduct a second sets of simulation study to explore its thermal and energy saving performance in buildings.  

Comment 1.3 Table 2: Thermophysical properties of aerogel render is missing.

Response 1.3 : Thermophysical properties of aerogel render is provided in Table 2.

Comment 1.4 Is there any particular reason why you chose the Melbourne climate?

Response 1.4 : Firstly, Melbourne exhibits a temperate oceanic climate which has a high diurnal temperature swing. Temperate climatic zones are advantageous for PCM application because it allows complete melting/freezing cycle during summer and improves summertime thermal comfort. Secondly, almost 86% of the existing building stock in the state of Victoria (Melbourne is the capital city of Victoria) was built before the introduction of mandatory energy efficiency standard in 2006 and hence, are energy in-efficient. This study aims to develop a retrofitting strategy to insulate the existing building stock in Melbourne using PCM and aerogel. That is why Melbourne's climate was chosen.  

These two aspects are now discussed in section 1 (introduction) and section 2.2 (Building energy and thermal simulations) of the revised manuscript.

Comment 1.5: Since you did not perform the EnergyPlus validation as part of your study, it should be moved from section 2.2 Building Energy and Thermal Simulations to 1. Introduction section (or perhaps reorganise the Introduction section and add an additional Literature Review section).

Response 1.6: We respectfully disagree with the reviewer on this point. The reviewer rightly pointed out that we did not do model validation in this study and used a validated model from our previously published research. We think that the paragraph about model validation is more appropriate in section 2.2 as we are discussing the model development here. Moving it to the introduction section will be confusing for the readers and compromise the coherency of this manuscript.

Comment 1.6 Table 4: It is not clear from the description what you mean by "Applying render on the outer/inner parts of the wall". Which walls? External walls? Internal walls?

Response 1.6 : The term “Applying render on the outer/inner parts of the wall” has been clarified as “applying render on the outer part of the exterior wall and inner part of the exterior and interior wall”.

Comment 1.7 Reader must carefully compare Table 4 with Table 5 to get the whole picture. Perhaps you could combine these two tables into a single table.

Response 1.7 :  Table 4 and table 5 are now combined into a single table 4 to present the simulated retrofit strategies.

Comment 1.8: Table 5 - ceiling insulation? I do not see its properties in Table 2. What is the construction detail of the ceiling? Is it a concrete slab or a wooden frame ceiling? Not clear... If it is a wooden frame ceiling, of course there is insulation between the wood rafters. That should be baseline case, not the retrofit case.
It is also not clear what the construction detail of external wall is. From Table 5, one might conclude that External wall is just brick veneer and plasterboard. This type of construction usually also consists of an air cavity and insulation, as you have model it in another publication https://www.sciencedirect.com/science/article/abs/pii/S0378778814003405

Response 1.8: Properties of ceiling insulation is now added in Table 1 (previously table 2). The construction detail of the ceiling for different retrofit cases is shown in Table 3. Moreover, Figure 7 and Table 3, reveal construction detail of ceiling, external walls and internal walls

This is a wooden frame house. We considered a house without any insulation in the ceiling and wall as the baseline case because this study is about retrofitting existing energy inefficient house. Ceiling insulation is easy, just need to put the insulation between the ceiling joists or wood rafter depending the house type. However, insulating the existing wall is the hard part because if you want to put traditional insulation material, you need to remove the plaster board, apply insulation and put a new plasterboard. This may not be practical in many cases. That is why, in this project, we are investigating an external insulating render. This discussion is now included in section 2.3,

In terms of external walls, a number of the old houses are made of brick and plasterboard without any insulation on the external walls. That is why in our baseline model insulation was not considered. We could have air cavity in our baseline model. However, it will not make any difference to the findings of this study. Because typical air cavity only adds 0.15 m2k/W resistance to the wall which is very insignificant compared to the wall insulation and insulated render that has been used in this study.

Comment 1.9: Please explain why you placed your PCM on the outer side of external wall (Cases 4 - 7)? Isn't the main point to locate PCM on the side of the conditioned zone so that it can respond to temperature changes of the indoor air rather than the external air? In Cases 4 - 7, placing PCM on the outer side of external wall it is exposed to the temperature changes of the outside air.
In Cases 10-13, on the other hand, you have placed PCM on the inner side of external wall (on the side of the conditioned zone). Now, cases e.g. 4 and 10 cannot be compared to study the effect of retrofitting the internal walls as well.

Response 1.9 : Mostly PCM panels and Form shape stable PCM (FSSPCM) composite integrated mortars are applied close to condition space for effective regulation of indoor temperature but it faces many technical, social, and economic challenges. For example, it suffers from incomplete solidification at night due to higher indoor temperatures. Although night ventilation can help to solidify the PCM, it is not always feasible in a residential building due to security, privacy and comfort issues.

Several studies investigated the application of high conductive coating such as, graphene, carbon nano tubes, graphite, and nano metallic powder to increase melting and solidification rates. However, highly conductive coating is not recommended due to structural integrity, safety concern and increased heating load in winter. Moreover, in summer, the absorbed heat by PCM during the daytime would be released to indoor at night which may exacerbate the night-time overheating issue

Practically, applying PCM panel and aerogel render on inner part of the external wall  and internal walls requires displacement of the occupants and their essentials which is one of the main hindrances to building energy retrofitting. Therefore, there is need to develop a highly insulating and heat storing render that can be easily applied on exterior side of wall without disturbing occupancy.

To allow direct comparison amongst different cases, we have kept the total amount of PCM constant in all cases. This was done by varying the thickness of PCM layer according to the application surface area. Hence, case 4 can be compared with case 10 because they were simulated considering the same quantity of PCM. This has been mentioned in section 2.3 of the manuscript.

Comment 1.10 Section 2.5.1 Indoor heat stress risk and thermal discomfort: how did you determine the wet bulb temperature?

Response 1.10: The EnergyPlus model by default calculates the dry-bulb temperature, relative humidity and barometric pressure of each zone at every time step during the simulation. The wet-bulb temperature was calculated using an advanced functionality of EnergyPlus known as EMS application which uses dry-bulb temperature, relative humidity and barometric pressure as input. This application has various psychometric functions which can be used to calculate user-defined output. One such function is the function for calculating Wet-bulb temperature as mentioned below:

 Wet-bulb temperature =  @TwbFnTdbWPb

Where,

@ TwbFn is the Wetbulb temperature function ( °C)

Input 1: Tdb,  Drybulb temperature( °C)

Input 2:  W, Humidity ratio (kgWater/kgDryAir)

Input 3: Pb, Barometric pressure (Pa)

In summary, we have written and this function in the EnergyPlus model via EMS application at every time step and calculated the corresponding wet-bulb temperature.   

Comment 1.11: Is there a specific reason why your diagrams are not visually consistent throughout the manuscript? For example, compare Figure 12 or Figure 16 with Figures 8, 9, 11, etc.

Response 1.11: All diagrams are redrawn with the same visual consistency.

Comment 1.12: This research is promising, but certain aspects need to be critically checked, well argued, and thoroughly revised. 

Response 1.12: We thank the reviewer for the encouraging and constructive comments . We have carefully addressed all the comments and thoroughly revised the manuscript as suggested.

Reviewer 2 Report

This research studies the performance of PCM integrated aerogel render in terms of heat stress, energy savings, peak cooling, emission, and lifecycle cost. a typical single-story Australian house was used as a case study. 
Concentrate on relevant information that the reader needs to understand your model. I have marked the most important things as follows:
The English language requires improvements. Please read the text carefully and search for typos ("a" "," "the", Capital letter, space between word, etc). For example, I fixed some part of the abstract as following that the authors could find out what I mean:
"The results showed that applying aerogel render and PCM on the outer side of the walls and installing PCM and ceilings insulation is the best option considering all performance indicators. "
The abstract is too long! An appealing abstract should contain:
1)      What has been done?
2)      What are the main findings?
in the introduction, it is better to first have a quick review to other multi-objective studies such as "A Multigeneration Cascade System Using Ground-source Energy with Cold Recovery: 3E Analyses and Multi-objective Optimization" and "Performance analysis of integrated solar heat pump VRF system for the low energy building in Mediterranean island" and "Application of hybrid systems in solution of low power generation at hot seasons for micro hydro systems" and EnrgyPlus using studies such as: "Thermal Performance of Electrochromic Smart Window with Nanocomposite Structure Under Different Climates In Iran" and "Analysis of energy consumption improvements of a zero-energy building in a humid mountainous area" to compare it with your method. There are some other previous related works in literature, which are good to read.
Please detail the abstract with the main conclusion in 1-2 sentences. 
"simulated in EnergyPlus" What are the other feasible alternatives? What are the advantages of adopting this particular metric over others in this case? How will this affect the results? More details should be furnished.
"the integration of PCM to improve thermal storage may have an adverse impact" This is strong and contentious statement without proof? (You have not made such a case based on literature) 
"Australian houses that were used to develop the house energy accreditation system in Australia" If this is your own conclusion, it is out of place here in same paragraph with statement, And should probably be part of the next paragraph that motives the reason for your research?  
"HVAC cases as mentioned in section 3.1." Highlighted sentence?
Color figures have been formatted so they are unclear when printed in black and white.
The ECO measures are not clear. Please use a clear definition.

Author Response

Reviewer 2

Comment 2.1 This research studies the performance of PCM integrated aerogel render in terms of heat stress, energy savings, peak cooling, emission, and lifecycle cost. a typical single-story Australian house was used as a case study. 
Concentrate on relevant information that the reader needs to understand your model. I have marked the most important things as follows:
The English language requires improvements. Please read the text carefully and search for typos ("a" "," "the", Capital letter, space between word, etc). For example, I fixed some part of the abstract as following that the authors could find out what I mean:
"The results showed that applying aerogel render and PCM on the outer side of the walls and installing PCM and ceilings insulation is the best option considering all performance indicators. "
The abstract is too long! An appealing abstract should contain:
1)      What has been done?
2)      What are the main findings?

Response 2.1: We thank the reviewer for the critical review and constructive comments. We have thoroughly revised the manuscript. The abstract has been re-written to highlight the research aim, methods and key findings.

Comment 2.2: In the introduction, it is better to first have a quick review to other multi-objective studies such as "A Multigeneration Cascade System Using Ground-source Energy with Cold Recovery: 3E Analyses and Multi-objective Optimization" and "Performance analysis of integrated solar heat pump VRF system for the low energy building in Mediterranean island" and "Application of hybrid systems in solution of low power generation at hot seasons for micro hydro systems" and EnrgyPlus using studies such as: "Thermal Performance of Electrochromic Smart Window with Nanocomposite Structure Under Different Climates In Iran" and "Analysis of energy consumption improvements of a zero-energy building in a humid mountainous area" to compare it with your method. There are some other previous related works in literature, which are good to read.

Response 2.2: Thank you very much for the suggestion. Please accept our apology for missing those references in the first place. The suggested references are now added in the introduction section.

Comment 2.3: Please detail the abstract with the main conclusion in 1-2 sentences. "simulated in EnergyPlus" What are the other feasible alternatives? What are the advantages of adopting this particular metric over others in this case? How will this affect the results? More details should be furnished.

Response 2.3: The abstract has been re-written to highlight the research aim, methods and key findings.

Comment 2.4: "the integration of PCM to improve thermal storage may have an adverse impact" This is strong and contentious statement without proof? (You have not made such a case based on literature) 

Response 2.4: We apologise for not being clear before. Previous studies showed that Integration of PCM reduces the compressive strength of construction materials. However, this statement is not related to the aim of our current study and is therefore deleted from the revised manuscript.

Comment 2.5: "Australian houses that were used to develop the house energy accreditation system in Australia" If this is your own conclusion, it is out of place here in same paragraph with statement, And should probably be part of the next paragraph that motives the reason for your research?  

"HVAC cases as mentioned in section 3.1." Highlighted sentence?

Response 2.5 We believe it belongs to case study building description section 2.1. This shows that the case study building actually represents a typical Australian house. We have now revised this section to add the following text:

The selected case study building is one of the eight representative Australian houses used to develop the nationwide house energy accreditation system (NatHERS) in Australia [1]. According to the Australian Building Code Board, the selected single-story house model is one of the two most typical representations of single-story detached houses in Australia. Approximately 72.9% of the Australian dwellings fall in the category of single-story detached house [2].”

The highlight was an unintentional mistake. It has now been removed. We apologise for any confusion.

Response 2.5: as mentioned second last line of last paragraph in section 3.1.

Comment 2.6: Color figures have been formatted so they are unclear when printed in black and white.
The ECO measures are not clear. Please use a clear definition.

Response 2.6: We apologise for the inconvenience caused by the unclear figures. All figures have been revised to improve readability.

[1]          C. f. I. Economics, "Proposal to revise energy efficiency requirements of the Building Code of Australia for residential buildings," Australian Building Codes Board, Canberra, 2009. [Online]. Available: https://www.abcb.gov.au/-/media/Files/Resources/Consultation/RIS-Energy-Efficiency-Residential-Building-Final-Decision-BCA-2010.pdf

[2]          ABS, "Census of population and housing," 2016. [Online]. Available: https://quickstats.censusdata.abs.gov.au/census_services/getproduct/census/2016/quickstat/036.

Reviewer 3 Report

This manuscript presented an investigation of retrofitting residential envelopes using PCM with aerogel render to reduce heat stress, energy use, CO2 emission, and construction cost. With 12 design scenarios. The research framework has scientific-method including case study selection and model validation. The study provides sufficient data analysis interesting results, which can be helpful for future implementation for typical Australian houses. The reliable discussion and conclusion were presented. There are few comments as following:

-       Table 3, the currency of AUD should be converted to USD.

-       Please provide the error range of the secondary validation

-       The manuscript should follow the journal template.

Author Response

These comments are very important for us. As much as we wanted to address these comments, we could not do so due tight submission deadline. This reviewer comments was not available to us when we received the major revision decision from the editor on 28th of August. We only saw this today just before submission and the revised manuscript submission is due today. The date stamp shows that this review was added on 7th September which is 11 days after we received the revision request.

Reviewer 4 Report

Generally, the authors need to focus on the objective of their paper and highlight it clearly. I see that the abstract and conclusion are not based on the core of the work, which is also needs further improvements in clarifying its necessity and value. My comments continue as follows;

A nomenclature page, including list of abbreviations should be given.

2. The novelty of this paper should also be well described and emphasized in the title, abstract, and conclusion. Please work on this and show us why this work is valuable.
The Abstract should contain answers to the following questions:
*       What problem was studied and why is it important?
*       What are the important results?
*       What conclusions can be drawn from the results?
*       What is the novelty of the work and where does it go beyond previous efforts in the literature?

2. The originality of the paper needs to be stated clearly and the most significative results.

3. Key terms should be defined as they arise in the paper, if necessary in relation to the literature.

4. The literature review should either be a conventional discourse review or a systematic review with the paper title reflecting the approach taken. Regardless, it would be useful to know the query terms and which online journal databases were used to identify the relevant literature.

5. Please summarise the literature review and exercise selectiveness in order to mainly support the main aim and objectives.

6. Results of the literature review may be presented in tabular format or using similar media to reduce the word count.

7. Present reasons for/statistical evidence of  single storey Australian house is ‘nationally representative’. Explain how the sample is random and not just chance. Include use of census data.

8. The conclusion should be kept succinct. Future work and impacts should be discussed elsewhere.

Author Response

Reviewer-4

Generally, the authors need to focus on the objective of their paper and highlight it clearly. I see that the abstract and conclusion are not based on the core of the work, which is also needs further improvements in clarifying its necessity and value. My comments continue as follows;

Comment 4.1: A nomenclature page, including list of abbreviations should be given.

Response 4.1: A nomenclature page has been added as per the suggestion after conclusion

Comment 4.2 The novelty of this paper should also be well described and emphasized in the title, abstract, and conclusion. Please work on this and show us why this work is valuable.
The Abstract should contain answers to the following questions:
*       What problem was studied and why is it important?
*       What are the important results?
*       What conclusions can be drawn from the results?
*       What is the novelty of the work and where does it go beyond previous efforts in the literature?

Response 4.2: The abstract has been re-written to highlight the research problem, research aim, methods and key findings. The novelty of this research has also been pointed out.

Comment 4.3. The originality of the paper needs to be stated clearly and the most significative results.

Response 4.3: The novelty of the paper has been presented clearly in the revised version.

Comments 4.4. Key terms should be defined as they arise in the paper, if necessary, in relation to the literature.

Response 4.4 The manuscript has been revised to include the definitions of all the key terms that can assist in understanding the topic and the work done

Comments 4.5: The literature review should either be a conventional discourse review or a ystematic review with the paper title reflecting the approach taken. Regardless, it would be useful to know the query terms and which online journal databases were used to identify the relevant literature.

Response 4.5: The literature review was done by manually extracting the information from the relevant journals and reports. The relevant journals and reports were obtained by searching the key words “Aerogel”, “Building energy retrofitting”, “Phase change materials” in the Scopus and “web of science” database.

Comment 4.6: Please summarise the literature review and exercise selectiveness in order to mainly support the main aim and objectives.

Response 4.6: We have thoroughly revised the literature review to highlight the research gap and support our research aims and objectives.

Comment 4.7: Results of the literature review may be presented in tabular format or using similar media to reduce the word count.

Response 4.7. We agree with the reviewer that the literature review is quite long. We have revised the literature review and deleted the irrelevant part to make it concise. The word count has been reduced considerably after revision. Therefore, a presentation of tabular format was not required.

Comment 4.8: Present reasons for/statistical evidence of  single storey Australian house is ‘nationally representative’. Explain how the sample is random and not just chance. Include use of census data.

Response 4.8: The selected case study building is one of the eight representative Australian houses used to develop the nationwide house energy accreditation system (NatHERS) in Australia [1]. According to the Australian Building Code Board, the selected single-story house model is one of the two most typical representations of single-story detached houses in Australia. Approximately 72.9% of the Australian dwellings fall in the category of single-story detached house [2].

Comment 4.9: The conclusion should be kept succinct. Future work and impacts should be discussed elsewhere.

Response 4.9: Conclusion has been revised and the unnecessary information has been deleted. The limitations and impacts have been moved to section 4.2.

Round 2

Reviewer 1 Report

I would like to thank the authors for addressing my main concerns from the review. The revised version of the manuscript appears to be good. I have only a few additional (minor) comments to make:

  • General comment: I would suggest using a different phrase throughout the manuscript, something different instead of "integrated". Something that makes it clear that PCM and aerogel are not "mixed"/encapsulated/etc. in your study. Maybe something more "neutral", like PCM combined with aerogel render.... or aerogel-based render coupled with PCM
  • Comment on Response 1.2: I know you mentioned this in Section 2.1, but I am missing in the Conclusions that you plan to use simulations from this study to develop and experimentally test PCM integrated aerogel render. It would be great to compare the results of the second set of simulations (PCM integrated aerogel render with properties known from experimental testing) with the results of this first modelling assumption (separate layers). I would encourage you to highlight this in the Conclusions section as a future research direction.
  • Comment on Response 1.3: Ok, but specify that it is aerogel render, not just "Render".
  • Comment on Response 1.10: You can end the explanation of wet bulb temperature with the sentence "....as EMS application which uses dry- bulb temperature, relative humidity and barometric pressure as input." No further explanation or formulas are required. It is sufficient to refer the reader to the EnergyPlus EMS application.
  • Conclusions: "The optimum thickness for PCM and aerogel render was 0.025m considering the emission, comfort, energy, and life cycle costs." ->State that this is for a case analysed (a typical Australian house in Melbourne climate) and not a general conclusion.

Author Response

I would like to thank the authors for addressing my main concerns from the review. The revised version of the manuscript appears to be good. I have only a few additional (minor) comments to make:

  • General comment: I would suggest using a different phrase throughout the manuscript, something different instead of "integrated". Something that makes it clear that PCM and aerogel are not "mixed"/encapsulated/etc. in your study. Maybe something more "neutral", like PCM combined with aerogel render.... or aerogel-based render coupled with PCM

Response: We thank the reviewer for this suggestion. To avoid any confusion, the suggested phrase “PCM combined with aerogel render” was used in relevant places throughout the manuscript instead of PCM integrated aerogel render

  • Comment on Response 1.2: I know you mentioned this in Section 2.1, but I am missing in the Conclusions that you plan to use simulations from this study to develop and experimentally test PCM integrated aerogel render. It would be great to compare the results of the second set of simulations (PCM integrated aerogel render with properties known from experimental testing) with the results of this first modelling assumption (separate layers). I would encourage you to highlight this in the Conclusions section as a future research direction.

Response: In conclusion, the potential future studies are given as follow.

“This study is the first step of a PCM integrated aerogel render development project. In the future, the findings from this simulation study will be used to develop a PCM integrated aerogel render. The thermal properties and performance of the developed render will be evaluated experimentally. Then the numerical model developed in this study will be updated to include the properties of PCM integrated aerogel render, and its performance will be compared against the results presented in this study where PCM and aerogel renders were assumed as separate layers.  Furthermore, the thermal performance of the PCM integrated aerogel render will be evaluated for different climate zones”

  • Comment on Response 1.3: Ok, but specify that it is aerogel render, not just "Render".

Response: Render is specified as aerogel render.

  • Comment on Response 1.10: You can end the explanation of wet bulb temperature with the sentence "....as EMS application which uses dry- bulb temperature, relative humidity and barometric pressure as input." No further explanation or formulas are required. It is sufficient to refer the reader to the EnergyPlus EMS application.

Response: Thank you. Further explanation of EMS is omitted in the revised manuscript as suggested. A reference to suggested EMS application document has been added.

  • Conclusions: "The optimum thickness for PCM and aerogel render was 0.025m considering the emission, comfort, energy, and life cycle costs." ->State that this is for a case analysed (a typical Australian house in Melbourne climate) and not a general conclusion.

Response: The highlighted sentence is re-written as “The optimum thickness for PCM and aerogel render was 0.025m considering the emission, comfort, energy, and life cycle costs for a typical Australian house in Melbourne climate”

Reviewer 2 Report

The paper has been modified based on the raised comments, and now, it could be accepted for publication.

Author Response

Thank you

Reviewer 4 Report

The authors have been addressed all changes accordingly and the paper is noteworthy to be published.

Author Response

Thank you